# Beyond Black-Box Advice: Learning-Augmented Algorithms for MDPs with Q-Value Predictions

**Tongxin Li**
School of Data Science
CUHK-SZ, China
litongxin@cuhk.edu.cn

**Yiheng Lin**
Computing + Mathematical Sciences
Caltech, USA
yihengl@caltech.edu

**Shaolei Ren**
Electrical & Computer Engineering
UC Riverside, USA
shaolei@ucr.edu

**Adam Wierman**
Computing + Mathematical Sciences
Caltech, USA
adamw@caltech.edu

## Abstract

We study the tradeoff between consistency and robustness in the context of a single-trajectory time-varying Markov Decision Process (MDP) with untrusted machine-learned advice. Our work departs from the typical approach of treating advice as coming from black-box sources by instead considering a setting where additional information about how the advice is generated is available. We prove a first-of-its-kind consistency and robustness tradeoff given Q-value advice under a general MDP model that includes both continuous and discrete state/action spaces. Our results highlight that utilizing Q-value advice enables dynamic pursuit of the better of machine-learned advice and a robust baseline, thus result in near-optimal performance guarantees, which provably improves what can be obtained solely with black-box advice.

## 1 Introduction

Machine-learned predictions and hand-crafted algorithmic advice are both crucial in online decision-making problems, driving a growing interest in *learning-augmented algorithms* [1, 2] that exploit the benefits of predictions to improve the performance for typical problem instances while bounding the worst-case performance [3, 4]. To this point, the study of learning-augmented algorithms has primarily viewed machine-learned advice as potentially untrusted information generated by black-box models. Yet, in many real-world problems, additional knowledge of the machine learning models used to produce advice/predictions is often available and can potentially improve the performance of learning-augmented algorithms.

A notable example that motivates our work is the problem of minimizing costs (or maximizing rewards) in a single-trajectory Markov Decision Process (MDP). More concretely, a value-based machine-learned policy $\widetilde{\pi}$ can be queried to provide suggested actions as advice to the agent at each step [5–7]. Typically, the suggested actions are chosen to minimize (or maximize, in case of rewards) estimated cost-to-go functions (known as Q-value predictions) based on the current state.

Naturally, in addition to suggested actions, the Q-value function itself can also provide additional information (e.g., the long-term impact of choosing a certain action) potentially useful to the design of a learning-augmented algorithm. Thus, this leads to two different designs for learning-augmented algorithms in MDPs: *black-box* algorithms and *grey-box* algorithms. A learning-augmented algorithm using $\widetilde{\pi}$ is black-box if $\widetilde{\pi}$ provides only the suggested action $\widetilde{u}$ to the learning-augmented algorithm,

37th Conference on Neural Information Processing Systems (NeurIPS 2023).

whereas it is value-based (a.k.a., grey-box) if $\widetilde{\pi}$ provides an estimate of the Q-value function $\widetilde{Q}$ (that also implicitly includes a suggested action $\widetilde{u}$ obtained by minimizing $\widetilde{Q}$) to the learning-augmented algorithm.

Value-based policies $\widetilde{\pi}$ often perform well empirically in stationary environments in practice [5, 6]. However, they may not have performance guarantees in all environments and can perform poorly at times due to a variety of factors, such as non-stationary environments [8–11], policy collapse [12], sample inefficiency [13], and/or when training data is biased [14]. As a consequence, such policies often are referred to as "untrusted advice" in the literature on learning-augmented algorithms, where the notion of "untrusted" highlights the lack of performance guarantees. In contrast, recent studies in competitive online control [15–21] have begun to focus on worst-case analysis and provide control policies $\overline{\pi}$ with strong performance guarantees even in adversarial settings, referred to as *robustness*, i.e., $\overline{\pi}$ provides "trusted advice." Typically, the goal of a learning-augmented online algorithm [1, 3] is to perform nearly as well as the untrusted advice when the machine learned policy performs well, a.k.a., achieve *consistency*, while also ensuring worst-case robustness. Combining the advice of an untrusted machine-learned policy $\widetilde{\pi}$ and a robust policy $\overline{\pi}$ naturally leads to a tradeoff between consistency and robustness. In this paper, we explore this tradeoff in a time-varying MDP setting and seek to answer the following key question for learning-augmented online algorithms:

*Can Q-value advice from an untrusted machine-learned policy, $\widetilde{\pi}$, in a **grey-box** scenario provide more benefits than the **black-box** action advice generated by $\widetilde{\pi}$ in the context of **consistency and robustness tradeoffs** for MDPs?*

## 1.1 Contributions

We answer the question above in the affirmative by presenting and analyzing a unified projection-based learning-augmented online algorithm (PROjection Pursuit policy, simplified as PROP in Algorithm 1) that combines action feedback from a trusted, robust policy $\overline{\pi}$ with an untrusted ML policy $\widetilde{\pi}$. In addition to offering a consistency and robustness tradeoff for MDPs with black-box advice, our work moves beyond the black-box setting. Importantly, by considering the grey-box setting, the design of PROP demonstrates that the *structural information* of the untrusted machine-learned advice can be leveraged to determine the trust parameters dynamically, which would otherwise be challenging (if not impossible) in a black-box setting. To our best knowledge, PROP is the first-of-its-kind learning-augmented algorithm that applies to general MDP models, which allow continuous or discrete state and action spaces.

Our main results characterize the tradeoff between consistency and robustness for both black-box and grey-box settings in terms of the ratio of expectations, RoE, built upon the traditional consistency and robustness metrics in [3, 22, 23, 4] for the competitive ratio. We show in Theorem 5.2 that for the black-box setting, PROP is $(1 + \mathcal{O}((1 - \lambda)\gamma))$-consistent and $(\text{ROB} + \mathcal{O}(\lambda\gamma))$-robust where $0 \leq \lambda \leq 1$ is a hyper-parameter. Moreover, for the black-box setting, PROP cannot be both $(1 + o(\lambda\gamma))$-consistent and $(\text{ROB} + o((1 - \lambda)\gamma))$-robust for any $0 \leq \lambda \leq 1$ where $\gamma$ is the diameter of the action space. In sharp contrast, by using a careful design of a robustness budget parameter in PROP with Q-value advice (grey-box setting), PROP is 1-consistent and $(\text{ROB} + o(1))$-robust.

Our result highlights the benefits of exploiting the additional information informed by the estimated Q-value functions, showing that the ratio of expectations can approach the better of the two policies $\widetilde{\pi}$ and $\overline{\pi}$ for any single-trajectory time-varying, and even possibly adversarial environments — if the value-based policy $\widetilde{\pi}$ is near-optimal, then the worst-case RoE(PROP) can approach 1 as governed by a consistency parameter; otherwise, RoE(PROP) can be bounded by the ratio of expectations of $\overline{\pi}$ subject to an additive term $o(1)$ that decreases when the time horizon $T$ increases.

A key technical contribution of our work is to provide the first quantitative characterization of the consistency and robustness tradeoff for a learning-augmented algorithm (PROP) in a general MDP model, under both standard black-box and novel grey-box settings. Importantly, PROP is able to leverage a broad class of robust policies, called *Wasserstein robust* policies, which generalize the well-known contraction principles that are satisfied by various robust policies [24] and have been used to derive regrets for online control [19, 25]. A few concrete examples of Wasserstein robust policies applicable for PROP are provided in Table 1 (Section 3.1).

## 1.2 Related Work

**Learning-Augmented Algorithms with Black-Box Advice.** The concept of integrating black-box machine-learned guidance into online algorithms was initially introduced by [26]. [3] coined terms "robustness" and "consistency" with formal mathematical definitions based on the competitive ratio. Over the past few years, the consistency and robustness approach has gained widespread popularity and has been utilized to design online algorithms with black-box advice for various applications, including ski rental [3, 22, 23], caching [27–29], bipartite matching [30], online covering [31, 32], convex body chasing [4], nonlinear quadratic control [33]. The prior studies on learning-enhanced algorithms have mainly focused on creating meta-strategies that combine online algorithms with black-box predictions, and typically require manual setting of a trust hyper-parameter to balance consistency and robustness. A more recent learning-augmented algorithm in [33] investigated the balance between competitiveness and stability in nonlinear control in a black-box setting. However, this work limits the robust policy to a linear quadratic regulator and does not provide a theoretical basis for the selection of the trust parameters. [34] generalized the black-box advice setting by considering distributional advice.

**Online Control and Optimization with Structural Information.** Despite the lack of a systematic analysis, recent studies have explored the usage of structural information in online control and optimization problems. Closest to our work, [7] considered a related setting where the Q-value function is available as advice, and shows that such information can be utilized to reduce regret in a tabular MDP model. In contrast, our analysis applies to more general models that allow continuous state/action spaces. In [17], the dynamical model and the predictions of disturbances in a linear control system are shown to be useful in achieving a near-optimal consistency and robustness tradeoff. The predictive optimization problem solved by MPC [35, 36, 16, 37] can be regarded as a special realization of grey-box advice, where an approximated cost-to-go function is constructed from structural information that includes the (predicted) dynamical model, costs, and disturbances.

**MDP with External Feedback.** Feedback from external sources such as control baselines [38, 39], visual explanations [40], and human experts [41–43] is often available in MDP. This external feedback can be beneficial for various purposes, such as ensuring safety [44], reducing variance [38], training human-like chatbots [41], and enhancing overall trustworthiness [45], among others. The use of control priors has been proposed by [38] as a way to guarantee the Lyapunov stability of the training process in reinforcement learning. They used the Temporal-Difference method to tune a coefficient that combines a RL policy and a control prior, but without providing a theoretical foundation. Another related area is transfer learning in RL, where external Q-value advice from previous tasks can be adapted and utilized in new tasks. Previous research has shown that this approach can outperform an agnostic initialization of Q, but these results are solely based on empirical observations and lack theoretical support [46–48].

## 2 Problem Setting

We consider a finite-horizon, single-trajectory, time-varying MDP with $T$ discrete time steps. The state space $\mathcal{X}$ is a subset of a normed vector space embedded with a norm $\|\cdot\|_{\mathcal{X}}$. The actions are chosen from a convex and compact set $\mathcal{U}$ in a normed vector space characterized by some norm $\|\cdot\|_{\mathcal{U}}$. Notably, $\mathcal{U}$ can represent either continuous actions or the probability distributions used when choosing actions from a finite set.[1] The diameter of the action space $\mathcal{U}$ is denoted by $\gamma := \max_{u \in \mathcal{U}} \|u\|_{\mathcal{U}}$. Denote $[T] := \{0, \ldots, T-1\}$. For each time step $t \in [T]$, let $P_t : \mathcal{X} \times \mathcal{U} \to \mathcal{P}_{\mathcal{X}}$ be the transition probability, where $\mathcal{P}_{\mathcal{X}}$ is a set of probability measures on $\mathcal{X}$. We consider time-varying costs $c_t : \mathcal{X} \times \mathcal{U} \to \mathbb{R}_+$, while rewards can be treated similarly by adding a negative sign. An initial state $x_0 \in \mathcal{X}$ is fixed. This MDP model is compactly represented by $\mathsf{MDP}(\mathcal{X}, \mathcal{U}, T, P, c)$.

The goal of a policy in this MDP setting is to minimize the total cost over all $T$ steps. The policy agent has no access to the full MDP. At each time step $t \in [T]$, only the incurred cost value $c_t(x_t, u_t)$ and the next state $x_{t+1} \sim P_t(\cdot | x_t, u_t)$ are revealed to the agent after playing an action $u_t \in \mathcal{U}$. We denote a policy by $\pi = (\pi_t : t \in [T])$ where each $\pi_t : \mathcal{X} \to \mathcal{U}$ chooses an action $u_t$ when observing

---

[1]The action space $\mathcal{U}$ is assumed to be a continuous, convex, and compact set for more generality. When the actions are discrete, $\mathcal{U}$ can be defined as the set of all probability distributions on a finite action space. We relegate the detailed discussions in Appendix A.2 and F.

$x_t$ at step $t \in [T]$. Note that our results can be generalized to the setting when $\pi_t$ is stochastic and outputs a probability distribution on $\mathcal{U}$. Given $\mathsf{MDP}(\mathcal{X}, \mathcal{U}, T, P, c)$, we consider an optimization with time-varying costs and transition dynamics. Thus, our goal is to find a policy $\pi$ that minimizes the following expected total cost:

$$J(\pi) \coloneqq \mathbb{E}_{P,\pi} \Big[ \sum_{t \in [T]} c_t \left( x_t, \pi_t(x_t) \right) \Big] \qquad (1)$$

where the randomness in $\mathbb{E}_{P,\pi}$ is from the transition dynamics $P = (P_t : t \in [T])$ and the policy $\pi = (\pi_t : t \in [T])$. We focus our analysis on the expected dynamic regret and the ratio of expectations, defined below, as the performance metrics for our policy design.

**Definition 1** (Expected dynamic regret). *Given* $\mathsf{MDP}(\mathcal{X}, \mathcal{U}, T, P, c)$*, the (expected) dynamic regret of a policy* $\pi = (\pi_t : t \in [T])$ *is defined as the difference between the expected cost induced by the policy* $\pi$*,* $J(\pi)$ *in (1), and the optimal expected cost* $J^\star \coloneqq \inf_\pi J(\pi)$*, i.e.,* $\mathsf{DR}(\pi) \coloneqq J(\pi) - J^\star$*.*

Dynamic regret is a more general (and often more challenging to analyze) measure than classical static regret, which has been mostly used for stationary environments [49, 50]. The following definition of the ratio of expectations [51, 52] will be used as an alternative performance metric in our main results.

**Definition 2** (Ratio of expectations). *Given* $\mathsf{MDP}(\mathcal{X}, \mathcal{U}, T, P, c)$*, the ratio of expectations of a policy* $\pi = (\pi_t : t \in [T])$ *is defined as* $\mathsf{RoE}(\pi) \coloneqq J(\pi)/J^\star$ *where* $J(\pi)$ *and* $J^\star$ *are the same as in Definition 1.*

Dynamic regret and the ratio of expectations defined above also depend on the error of the untrusted ML advice; we make this more explicit in Section 3.2. Next, we state the following continuity assumption, which is standard in MDPs with continuous action and state spaces [53–55]. Note that our analysis can be readily adapted to general Hölder continuous costs with minimal modifications.

**Assumption 1** (Lipschitz costs). *For any time step* $t \in [T]$*, the cost function* $c_t : \mathcal{X} \times \mathcal{U} \to \mathbb{R}_+$ *is Lipschitz continuous with a Lipschitz constant* $L_C < \infty$*, i.e., for any* $t \in [T]$*,* $|c_t(x, u) - c_t(x', u')| \le L_C \left( \|x - x'\|_\mathcal{X} + \|u - u'\|_\mathcal{U} \right)$*. Moreover,* $0 < c_t(x, u) < \infty$ *for all* $t \in [T]$*,* $x \in \mathcal{X}$*, and* $u \in \mathcal{U}$*.*

## 3 Consistency and Robustness in MDPs

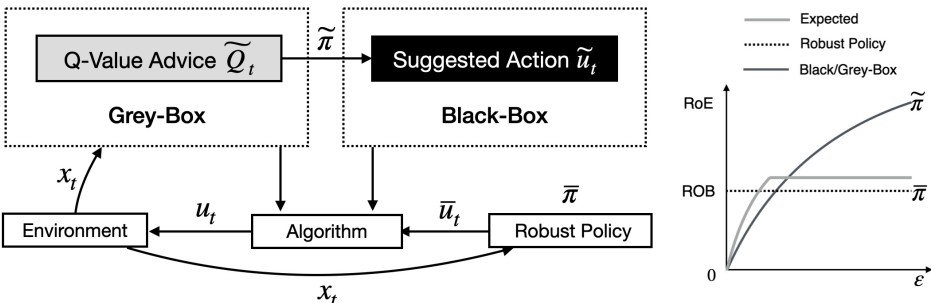

Figure 1: *Left*: Overview of settings in our problem. *Right*: consistency and robustness tradeoff, with RoE and $\varepsilon$ defined in Definition 2 and Equation (4).

Our objective is to achieve a balance between the worst-case guarantees on cost minimization in terms of dynamic regret provided by a robust policy, $\overline{\pi}$, and the average-case performance of a valued-based policy, $\widetilde{\pi}$, in the context of $\mathsf{MDP}(\mathcal{X}, \mathcal{U}, T, P, c)$. In particular, we denote by $\mathsf{ROB} \ge 1$ a ratio of expectation bound of the robust policy $\overline{\pi}$ such that the worst case $\mathsf{RoE}(\overline{\pi}) \le \mathsf{ROB}$. In the learning-augmented algorithms literature, these two goals are referred to as consistency and robustness [3, 1]. Informally, robustness refers to the goal of ensuring worst-case guarantees on cost minimization comparable to those provided by $\overline{\pi}$ and consistency refers to ensuring performance nearly as good as $\widetilde{\pi}$ when $\widetilde{\pi}$ performs well (e.g., when the instance is not adversarial). Learning-augmented algorithms seek to achieve consistency and robustness by combining $\overline{\pi}$ and $\widetilde{\pi}$, as illustrated in Figure 1.

Table 1: Examples of models covered in this paper and the associated control baselines. For the right column, bounds on the ratio of expectations RoE are exemplified, where ROB is defined in Section 3 and $\mathcal{O}$ omits inessential constants.

| Model | Robust Baseline $\overline{\pi}$ | RoE |
|---|---|---|
| Time-varying MDP (Our General Model) | Wasserstein Robust Policy (Definition 3) | ROB |
| Discrete MDP (Appendix A.2) | Any Policy that Induced a Regular Markov Chain | — |
| Time-Varying LQR (Appendix A.1) | MPC with Robust Predictions (Algorithm 3) | $\mathcal{O}(1)$ |

Our focus in this work is to design robust and consistent algorithms for two types of advice: black-box advice and grey-box advice. The type of advice that is nearly always the focus in the learning-augmented algorithm literature is black-box advice — only providing a suggested action $\widetilde{u}_t$ without additional information. In contrast, on top of the action $\widetilde{u}_t$, grey-box advice can also reveal the internal state of the learning algorithm, e.g., the Q-value $\widetilde{Q}_t$ in our setting. This contrast is illustrated in Figure 1.

Compared to black-box advice, grey-box advice has received much less attention in the literature, despite its potential to improve tradeoffs between consistency and robustness as recently shown in [34, 17]. Nonetheless, the extra information on top of the suggested action in a grey-box setting potentially allows the learning-augmented algorithm to make a better-informed decision based on the advice, thus achieving a better tradeoff between consistency and robustness than otherwise possible.

In the remainder of this section, we discuss the robustness properties for the algorithms we consider in our learning-augmented framework (Section 3.1), and introduce the notions of consistency in our grey-box and black-box models in Section 3.2.

## 3.1 Locally Wasserstein-Robust Policies

We begin with constructing a novel notion of robustness for our learning-augmented framework based on the Wasserstein distance as follows. Denote the robust policy by $\overline{\pi} := (\overline{\pi}_t : t \in [T])$, where each $\overline{\pi}_t$ maps a system state to a deterministic action (or a probability of actions in the stochastic setting). Denote by $\rho_{t_1:t_2}(\rho)$ the joint distribution of the state-action pair $(x_t, u_t) \in \mathcal{X} \times \mathcal{U}$ at time $t_2 \in [T]$ when implementing the baselines $\overline{\pi}_{t_1}, \ldots, \overline{\pi}_{t_2}$ consecutively with an initial state-action distribution $\rho$. We use $\| \cdot \|_{\mathcal{X} \times \mathcal{U}} := \| \cdot \|_{\mathcal{X}} + \| \cdot \|_{\mathcal{U}}$ as the included norm for the product space $\mathcal{X} \times \mathcal{U}$. Let $W_p(\mu, \nu)$ denote the Wasserstein $p$-distance between distributions $\mu$ and $\nu$ whose support set is $\mathcal{X} \times \mathcal{U}$:

$$W_p(\mu, \nu) := \left( \inf_{J \in \mathcal{J}(\mu, \nu)} \int \|(x,u) - (x',u')\|_{\mathcal{X} \times \mathcal{U}}^p \, \mathrm{d}J\left((x,u),(x',u')\right) \right)^{1/p}$$

where $p \in [1, \infty)$ and $\mathcal{J}(\mu, \nu)$ denotes a set of all joint distributions $J$ with a support set $\mathcal{X} \times \mathcal{U}$ that have marginals $\mu$ and $\nu$. Next, we define a robustness condition for our learning-augmented framework.

**Definition 3** ($r$-locally $p$-Wasserstein robustness)**.** *A policy $\overline{\pi} = (\pi_t : t \in [T])$ is $r$-**locally $p$-Wasserstein-robust** if for any $0 \leq t_1 \leq t_2 < T$ and any pair of state-action distributions $\rho, \rho'$ where the the $p$-Wasserstein distance between them is bounded by $W_p(\rho, \rho') \leq r$, for some radius $r > 0$, the following inequality holds:*

$$W_p\left(\rho_{t_1:t_2}(\rho), \rho_{t_1:t_2}(\rho')\right) \leq s(t_2 - t_1) W_p\left(\rho, \rho'\right) \tag{2}$$

*for some function $s : [T] \to \mathbb{R}_+$ satisfying $\sum_{t \in [T]} s(t) \leq C_s$ where $C_s > 0$ is a constant.*

Our robustness definition is naturally more relaxed than the usual contraction property in the control/optimization literature [25, 35] — if any two different state-action distributions converge exponentially with respect to the Wasserstein $p$-distance, then a policy $\overline{\pi}$ is $r$-*locally $p$-Wasserstein-robust*. This is illustrated in Figure 2. Note that, although the Wasserstein robustness in Definition 3 well captures a variety of distributional robustness metrics such as the total variation robustness defined on finite state/action spaces, it can also be further generalized to other metrics for probability distributions.

As shown in Appendix A (provided in the supplementary material), by establishing a connection between the Wasserstein distance and the total variation metric, any policy that induces a regular

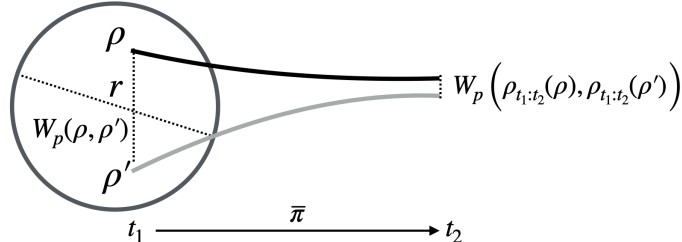

Figure 2: An illustration of an *r-locally p-Wasserstein-robust* policy.

Markov chain satisfies the fast mixing property and the state-action distribution will converge with respect to the total variation distance to a stationary distribution [56]. A more detailed discussion can be found in Appendix A.2. Moreover, the Wasserstein-robustness in Definition 3 includes a set of contraction properties in control theory as special cases. For example, for a locally Wasserstein-robust policy, if the transition kernel $P$ and the baseline policy $\overline{\pi}$ are deterministic, then the state-action distributions become point masses, reducing Definition 3 to a state-action perturbation bound in terms of the $\ell_2$-norm when implementing the policy $\overline{\pi}$ from different starting states [35, 19].

The connections discussed above highlight the existence of several well-known robust policies that satisfy Definition 3. Besides the case of discrete MDPs discussed in Appendix A.2, another prominent example is model predictive control (MPC), for which robustness follows from the results in [19] (see Appendix A.1 for details). The model assumption below will be useful in our main results.

**Assumption 2.** *There exists a $\gamma$-locally p-Wasserstein-robust baseline control policy (Definition 3) $\overline{\pi}$ for some $p \geq 1$, where $\gamma$ is the diameter of the action space $\mathcal{U}$.*

### 3.2 Consistency and Robustness for RoE

In parallel with the notation of "consistency and robustness" in the existing literature on learning-augmented algorithms [3, 1], we define a new metric of consistency and robustness in terms of RoE. To do so, we first introduce an optimal policy $\pi^\star$. Based on $\mathsf{MDP}(\mathcal{X}, \mathcal{U}, T, P, c)$, let $\pi_t^\star = (\pi_t^\star : t \in [T])$ denote the optimal policy at each time step $t \in [T]$, whose optimal Q-value function is

$$Q_t^\star(x, u) := \inf_\pi \mathbb{E}_{P,\pi} \left[ \sum_{\tau=t}^{T-1} c_\tau(x_\tau, u_\tau) \,\Big|\, x_t = x, u_t = u \right],$$

where $\mathbb{E}_{P,\pi}$ denotes an expectation with respect to the randomness of the trajectory $\{(x_t, u_t) : t \in [T]\}$ obtained by following a policy $\pi$ and the transition probability $P$ at each step $t \in [T]$. The Bellman optimality equations can then be expressed as

$$Q_t^\star(x, u) = (c_t + \mathbb{P}_t V_{t+1}^\star)(x, u), \qquad V_t^\star(x) = \inf_{v \in \mathcal{U}} Q_t^\star(x, v), \qquad V_t^\star(x) = 0 \qquad (3)$$

for all $(x, u) \in \mathcal{X} \times \mathcal{U}, t \in [T]$ and $t \in [T]$, where we write $(\mathbb{P}_t V^\star)(x, u) := \mathbb{E}_{x' \sim P_t(\cdot | x, u)}[V^\star(x')]$. This indicates that for each time step $t \in [T]$, $\pi_t^\star$ is the greedy policy with respect to its optimal Q-value functions $(Q_t^\star : t \in [T])$. Note that for any $t \in [T]$, $Q_t^\star(x, u) = 0$. Given this setup, the value-based policies $\widetilde{\pi} := (\widetilde{\pi}_t : t \in [T])$ take the following form. For any $t \in [T]$, a value-based policy $\widetilde{\pi}_t : \mathcal{X} \to \mathcal{U}$ produces an action $\widetilde{u}_t \in \arg\min_{v \in \mathcal{U}} \widetilde{Q}_t(x_t, v)$ by minimizing an estimate of the optimal Q-value function $\widetilde{Q}_t$.

We make the following assumption on the machine-learned untrusted policy $\widetilde{\pi}$ and the Q-value advice.

**Assumption 3.** *The machine-learned untrusted policy $\widetilde{\pi}$ is value-based. The Q-value advice $\widetilde{Q}_t : \mathcal{X} \times \mathcal{U} \to \mathbb{R}$ is Lipschitz continuous with respect to $u \in \mathcal{U}$ for any $x \in \mathcal{X}$, with a Lipschitz constant $L_Q$ for all $t \in [T]$. Moreover, $\widetilde{Q}_t(x, u) - Q_t^\star(x, u) = o(T)$ for all $(x, u) \in \mathcal{X} \times \mathcal{U}$ and $t \in [T]$.*

We can now define a consistency measure for Q-value advice $\widetilde{Q}_t$, which measures the error of the estimates of the Q-value functions due to approximation error and time-varying environments, etc. Let $p \in (0, \infty]$. Fix a sequence of distributions $\rho = (\rho_t : t \in [T])$ whose support set is $\mathcal{X} \times \mathcal{U}$ and let $\phi_t$ be the marginal distribution of $\rho_t$ on $\mathcal{X}$. We define a quantity representing the error of the Q-value

advice

$$\varepsilon(p,\rho) \coloneqq \sum_{t \in [T]} \left( \left\| \widetilde{Q}_t - Q_t^\star \right\|_{p,\rho_t} + \left\| \inf_{v \in \mathcal{U}} \widetilde{Q}_t - \inf_{v \in \mathcal{U}} Q_t^\star \right\|_{p,\phi_t} \right) \tag{4}$$

where $\| \cdot \|_{p,\rho} \coloneqq \left( \int |\cdot|^p \, d\rho \right)^{1/p}$ denotes the $L_{p,\rho}$-norm. A policy with Q-value functions $\{Q_t : t \in [T]\}$ is said to be $(\varepsilon, p, \rho)$-*consistent* if there exists an $\varepsilon$ satisfying (4). In addition, a policy is $(0, \infty)$-consistent if $\widetilde{Q}_t$ is a Lebesgue-measurable function for all $t \in [T]$ and $(\infty, \varepsilon)$-consistent if the $L_\infty$-norm satisfies $\sum_{t \in [T]} \|\widetilde{Q}_t - Q_t^\star\|_\infty \le \varepsilon$. The consistency error of a policy in (4) quantifies how the Q-value advice is close to optimal Q-value functions. It depends on various factors such the function approximation error or training error due to the distribution shift, and has a close connection to a rich literature on value function approximation [57–61]. The results in [59] generalized the worst-case $L_\infty$ guarantees to arbitrary $L_{p,\rho}$-norms under some mixing assumptions via policy iteration for a stationary Markov decision process (MDP) with a continuous state space and a discrete action space. Recently, approximation guarantees for the average case for parametric policy classes (such as a neural network) of value functions have started to appear [57, 58, 60]. These bounds are useful in lots of supervised machine learning methods such as classification and regression, whose bounds are typically given on the expected error under some distribution. These results exemplify richer instances of the consistency definition (see (4)) and a summary of these bounds can be found in [61].

Now, we are ready to introduce our definition of consistency and robustness with respect to the ratio of expectations, similar to the growing literature on learning-augmented algorithms [3, 22, 23, 4]. We write the ratio of expectations $\mathsf{RoE}(\varepsilon)$ of a policy $\pi$ as a function of the Q-value advice error $\varepsilon$ in terms of the $L_\infty$ norm, defined in (4).

**Definition 4** (Consistency and Robustness). *An algorithm $\pi$ is said to be k-**consistent** if its worst-case (with respect to the MDP model $\mathsf{MDP}(\mathcal{X}, \mathcal{U}, T, P, c)$) ratio of expectations satisfies $\mathsf{RoE}(\varepsilon) \le k$ for $\varepsilon = 0$. On the other hand, it is l-**robust** if $\mathsf{RoE}(\varepsilon) \le l$ for any $\varepsilon > 0$.*

## 4 The Projection Pursuit Policy (PROP)

In this section we introduce our proposed algorithm (Algorithm 1), which achieves near-optimal consistency while bounding the robustness by leveraging a robust baseline (Section 3.1) in combination with value-based advice (Section 3.2). A key challenge in the design is how to exploit the benefits of good value-based advice while avoiding following it too closely when it performs poorly. To address this challenge, we propose to judiciously project the value-based advice into a neighborhood of the robust baseline. By doing so, the actions we choose can follow the value-based advice for consistency while staying close to the robust baseline for robustness. More specifically, at each step $t \in [T]$, we choose $u_t = \mathrm{Proj}_{\overline{\mathcal{U}}_t}(\widetilde{u}_t)$ where a projection operator $\mathrm{Proj}_{\overline{\mathcal{U}}_t}(\cdot) : \mathcal{U} \to \mathcal{U}$ is defined as

$$\mathrm{Proj}_{\overline{\mathcal{U}}_t}(u) \coloneqq \underset{v \in \mathcal{U}}{\arg\min} \, \|u - v\|_{\mathcal{U}} \text{ subject to } \|v - \overline{\pi}_t(x_t)\|_{\mathcal{U}} \le R_t, \tag{5}$$

corresponding to the projection of $u$ onto a ball $\overline{\mathcal{U}}_t \coloneqq \{u \in \mathcal{U} : \|u - \overline{\pi}_t(x_t)\|_{\mathcal{U}} \le R_t\}$. Note that when the optimal solution of (5) is not unique, we choose the one on the same line with $\overline{\pi}_t(x_t) - u$.

The PROjection Pursuit policy, abbreviated as PROP, can be described as follows. For a time step $t \in [T]$, let $\widetilde{\pi}_t : \mathcal{X} \to \mathcal{U}$ denote a policy that chooses an action $\widetilde{u}_t$ (arbitrarily choose one if there are multiple minimizers of $\widetilde{Q}_t$), given the current system state $x_t$ at time $t \in [T]$ and step $t \in [T]$. An action $u_t = \mathrm{Proj}_{\overline{\mathcal{U}}_t}(\widetilde{u}_t(x_t))$ is selected by projecting the machine-learned action $\widetilde{u}_t(x_t)$ onto a norm ball $\overline{\mathcal{U}}_t$ defined by the robust policy $\overline{\pi}$ given a radius $R_t \ge 0$. Finally, PROP applies to both black-box and grey-box settings (which differ from each other in terms of how the radius $R_t$ is decided). The results under both settings are provided in Section 5, revealing a tradeoff between consistency and robustness.

The radii $(R_t : t \in [T])$ can be interpreted as *robustness budgets* and are key design parameters that determine the consistency and robustness tradeoff. Intuitively, the robustness budgets reflect the trustworthiness on the value-based policy $\widetilde{\pi}$ — the larger budgets, the more trustworthiness and hence the more freedom for PROP to follow $\widetilde{\pi}$. How the robustness budget is chosen differentiates the grey-box setting from the black-box one.

---

**Algorithm 1 PRO**jection **P**ursuit **P**olicy (**PROP**)

---

**Initialize :** Untrusted policy $\widetilde{\pi} = (\widetilde{\pi}_t : t \in [T])$ and baseline policy $\overline{\pi} = (\overline{\pi}_t : t \in [T])$

---

**1 for** $t = 0, \ldots, T-1$ **do**

**2**     *//Implement black-box (Section 4.1) or grey-box (Section 4.2) procedures*

**3**     $(\widetilde{u}_t, R_t) \leftarrow \text{BLACK-BOX}(x_t)$ or $(\widetilde{u}_t, R_t) \leftarrow \text{GREY-BOX}(x_t)$

**4**     Set action $u_t = \text{Proj}_{\overline{\mathcal{U}}_t}(\widetilde{u}_t)$ where $\overline{\mathcal{U}}_t := \{u \in \mathcal{U} : \|u - \overline{\pi}_t(x_t)\|_{\mathcal{U}} \le R_t\}$

**5**     Sample next state $x_{t+1} \sim P_t(\cdot|x_t, u_t)$

**6 end**

---

## 4.1 Black-Box Setting

In the black-box setting, the only information provided by $\widetilde{\pi}$ is a suggested action $\widetilde{u}$ for the learning-augmented algorithm. Meanwhile, the robust policy $\overline{\pi}$ can also be queried to provide advice $\overline{u}$. Thus, without additional information, a natural way to utilize both $\widetilde{\pi}$ and $\overline{\pi}$ is to decide a projection radius at each time based on the how the obtained $\widetilde{u}$ and $\overline{u}$. More concretely, at each time $t \in [T]$, the robustness budget $R_t$ is chosen by the following BLACK-BOX Procedure, where we set $R_t = \lambda \eta_t$ with $\eta_t := \|\widetilde{u}_t - \overline{u}_t\|_{\mathcal{U}}$ representing the difference between the two advice measured in terms of the norm $\|\cdot\|_{\mathcal{U}}$ and $0 \le \lambda \le 1$ being a tradeoff hyper-parameter that measures the trustworthiness on the machine-learned advice. The choice of $R_t = \lambda \eta_t$ can be explained as follows. The value of $\eta_t$ indicates the intrinsic discrepancy between the robust advice and the machine-learned untrusted advice — the larger discrepancy, the more difficult to achieve good consistency and robustness simultaneously. Given a robust policy and an untrusted policy, by setting a larger $\lambda$, we allow the actual action to deviate more from the robust advice and to follow the untrusted advice more closely, and vice versa. $\lambda$ is a crucial hyper-parameter that can be pre-determined to yield a desired consistency and robustness tradeoff. The computation of $R_t$ is summarized in Procedure 1 below.

---

**Procedure 1** BLACK-BOX Procedure at $t \in [T]$ (Input: state $x_t$ and hyper-parameter $0 \le \lambda \le 1$)

---

Implement $\widetilde{\pi}_t$ and $\overline{\pi}_t$ to obtain $\widetilde{u}_t$ and $\overline{u}_t$, respectively.

Set robustness budget $R_t = \lambda \eta_t$ where $\eta_t := \|\widetilde{u}_t - \overline{u}_t\|_{\mathcal{U}}$; Return $(\widetilde{u}_t, R_t)$

---

## 4.2 Grey-Box Setting

In the grey-box setting, along with the suggested action $\widetilde{u}$, the value-based untrusted policy $\widetilde{\pi}$ also provides an estimate of the Q-value function $\widetilde{Q}$ that indicates the long-term cost impact of an action. To utilize such additional information informed by $\widetilde{Q}_t$ at each time $t \in [T]$, we propose a novel algorithm that dynamically adjusts the budget $R_t$ to further improve the consistency and robustness tradeoff. More concretely, let us consider the Temporal-Difference (TD) error $\text{TD}_t = c_{t-1} + \mathbb{P}_{t-1}\widetilde{V}_t - \widetilde{Q}_{t-1}$. Intuitively, if a non-zero TD-error is observed, the budget $R_t$ needs to be decreased so as to minimize the impact of the learning error. However, the exact TD-error is difficult to compute in practice, since it requires complete knowledge of the transition kernels $(P_t : t \in [T])$. To address this challenge, we use the following estimated TD-error based on previous trajectories:

$$\delta_t(x_t, x_{t-1}, u_{t-1}) := c_{t-1}(x_{t-1}, u_{t-1}) + \inf_{v \in \mathcal{U}} \widetilde{Q}_t(x_t, v) - \widetilde{Q}_{t-1}(x_{t-1}, u_{t-1}). \qquad (6)$$

Denote by $\beta > 0$ a hyper-parameter. Based on the estimated TD-error in (6), the *robustness budget* in Algorithm 1 is set as

$$R_t := \left[ \underbrace{\|\widetilde{\pi}_t(x_t) - \overline{\pi}_t(x_t)\|_{\mathcal{U}}}_{\text{Decision Discrepancy } \eta_t} - \frac{\beta}{L_Q} \sum_{s=1}^{t} \underbrace{\delta_s(x_s, x_{s-1}, u_{s-1})}_{\text{Approximate TD-Error}} \right]^+, \qquad (7)$$

which constitutes two terms. The first term $\eta_t := \|\widetilde{\pi}_t(x_t) - \overline{\pi}_t(x_t)\|$ measures the *decision discrepancy* between the untrusted policy $\widetilde{\pi}$ and the baseline policy $\overline{\pi}$, which normalizes the total budget, similar to the one used in the black-box setting in Procedure 1. The second term is the approximate TD-error, which is normalized by the Lipschitz constant $L_Q$ of Q-value functions. With these terms defined, the GREY-BOX Procedure below first chooses a suggested action $\widetilde{u}_t$ by minimizing $\widetilde{Q}_t$ and then decides a robustness budget $R_t$ using (7).

**Procedure 2** GREY-BOX Procedure at $t \in [T]$ (Input: state $x_t$ and hyper-parameter $0 \le \beta \le 1$)

Obtain advice $\widetilde{Q}_t$ and $\widetilde{u}_t$ where $\widetilde{u}_t \in \arg\inf_{v \in \mathcal{U}} \widetilde{Q}_t(x_t, v)$
Implement $\overline{\pi}_t$ and obtain $\overline{u}_t$
Set robustness budget $R_t$ as (7); Return $(\widetilde{u}_t, R_t)$

## 5 Main Results

We now formally present the main results for both the black-box and grey-box settings. Our results not only quantify the tradeoffs between consistency and robustness formally stated in Definition 4 with respect to the ratio of expectations, but also emphasize a crucial role that additional information about the estimated Q-values plays toward improving the consistency and robustness tradeoff.

### 5.1 Black-Box Setting

In the existing learning-augmented algorithms, the untrusted machine-learned policy $\widetilde{\pi}$ is often treated as a black-box that generates action advice $\widetilde{u}_t$ at each time $t \in [T]$. Our first result is the following general dynamic regret bound for the black-box setting (Section 4.1). We utilize the Big-O notation, denoted as $\mathcal{O}(\cdot)$ and $o(\cdot)$ to disregard inessential constants.

**Theorem 5.1.** *Suppose the machine-learned policy $\widetilde{\pi}$ is $(\infty, \varepsilon)$-consistent. For any MDP model satisfying Assumption 1,2, and 3, the expected dynamic regret of* PROP *with the* BLACK-BOX *Procedure is bounded by* $\mathsf{DR}(\mathsf{PROP}) \le \min\{\mathcal{O}(\varepsilon) + \mathcal{O}((1 - \lambda)\gamma T), \mathcal{O}((\mathsf{ROB} + \lambda\gamma - 1)T)\}$ *where $\varepsilon$ is defined in (4), $\gamma$ is the diameter of the action space $\mathcal{U}$, $T$ is the length of the time horizon,* ROB *is the ratio of expectations of the robust baseline $\overline{\pi}$, and $0 \le \lambda \le 1$ is a hyper-parameter.*

When $\lambda$ increases, the actual action can deviate more from the robust policy, making the dynamic regret potentially closer to that of the value-based policy. While the regret bound in Theorem 5.1 clearly shows the role of $\lambda$ in terms of controlling how closely we follow the robust policy, the dynamic regret given a fixed $\lambda \in [0, 1]$ grows linearly in $\mathcal{O}(T)$. In fact, the linear growth of dynamic regret holds even if the black-box policy $\widetilde{\pi}$ is consistent, i.e., $\varepsilon$ is small. This can be explained by noting the lack of dynamically tuning $\lambda$ to follow the better of the two policies — even when one policy is nearly perfect, the actual action still always deviates from it due to the fixed choice of $\lambda$.

Consider any MDP model satisfying Assumptions 1,2, and 3. Following the classic definitions of consistency and robustness (see Definition 4), we summarize the following characterization of PROP, together with a negative result in Theorem 5.3. Proofs of Theorem 5.1, 5.2, and 5.3 are detailed in Appendix C.

**Theorem 5.2** (BLACK-BOX Consistency and Robustness)**.** PROP *with the* BLACK-BOX *Procedure is $(1 + \mathcal{O}((1 - \lambda)\gamma))$-consistent and $(\mathsf{ROB} + \mathcal{O}(\lambda\gamma))$-robust where $0 \le \lambda \le 1$ is a hyper-parameter.*

**Theorem 5.3** (BLACK-BOX Impossibility)**.** PROP *with the* BLACK-BOX *Procedure cannot be both $(1 + o((1 - \lambda)\gamma))$-consistent and $(\mathsf{ROB} + o(\lambda\gamma))$-robust for any $0 \le \lambda \le 1$.*

### 5.2 Grey-Box Setting

To overcome the impossibility result in the black-box setting, we dynamically tune the robustness budgets by tapping into additional information informed by the estimated Q-value functions using the GREY-BOX Procedure (Section 4.2). By setting the robustness budgets in (7), an analogous result of Theorem 5.1 is given in Appendix D, which leads to a dynamic regret bound of PROP in the grey-box setting (Theorem D.1 in Appendix D). Consider any MDP model satisfying Assumptions 1,2, and 3. Our main result below indicates that knowing more structural information about a black-box policy can indeed bring additional benefits in terms of the consistency and robustness tradeoff, even if the black-box policy is untrusted.

**Theorem 5.4** (GREY-BOX Consistency and Robustness)**.** PROP *with the* GREY-BOX *Procedure is 1-consistent and $(\mathsf{ROB} + o(1))$-robust for some $\beta > 0$.*

Theorem 5.3 implies that using the BLACK-BOX Procedure, PROP cannot be 1-consistent and $(\mathsf{ROB} + o(1))$-robust, while this can be achieved using the GREY-BOX Procedure. On one hand, this theorem validates the effectiveness of the PROP policy with value-based machine-learned advice that

may not be fully trusted. On the other hand, this sharp contrast between the black-box and grey-box settings reveals that having access to information of value function can improve the tradeoff between consistency and robustness (see Definition 4) non-trivially. A proof of Theorem 5.4 can be found in Appendix D. Applications of our main results are discussed in Appendix A.

## 6 Concluding Remarks

Our results contribute to the growing body of literature on learning-augmented algorithms for MDPs and highlight the importance of considering consistency and robustness in this context. In particular, we have shown that by utilizing the *structural information* of machine learning methods, it is possible to achieve improved performance over a black-box approach. The results demonstrate the potential benefits of utilizing value-based policies as advice; however, there remains room for future work in exploring other forms of structural information.

**Limitations and Future Work.** One limitation of our current work is the lack of analysis of more general forms of black-box procedures. Understanding and quantifying the available structural information in a more systematic way is another future direction that could lead to advances in the design of learning-augmented online algorithms and their applications in various domains.

## Acknowledgement

We would like to thank the anonymous reviewers for their helpful comments. This work was supported in part by the National Natural Science Foundation of China (NSFC) under grant No. 72301234, the Guangdong Key Lab of Mathematical Foundations for Artificial Intelligence, and the start-up funding UDF01002773 of CUHK-Shenzhen. Yiheng Lin was supported by the Caltech Kortschak Scholars program. Shaolei Ren was supported in part by the U.S. U.S. National Science Foundation (NSF) under grant CNS–1910208. Adam Wierman was supported in part by the U.S. NSF under grants CNS–2146814, CPS–2136197, CNS–2106403, NGSDI–2105648.

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
