# A  Application Examples

In this section, we delve deeper into the practical applications of our main results, which provide a general consistency and robustness tradeoff. By presenting concrete examples, we aim to demonstrate the versatility and relevance of our findings to various real-world problems and scenarios. We consider the settings summarized in Table 1. These examples illustrate how our results can be applied to optimize tradeoffs between consistency and robustness for more specific models that can be represented as a nonstationary MDP. Additionally, these examples highlight the significance of considering the tradeoff between consistency and robustness in the design and implementation of decision-making algorithms in the learning-augmented framework, and the impact of the structural information in the grey-box setting.

## A.1  MPC baseline in time-varying dynamical systems

The first application is an online optimal control problem, which is a special case of the general MDP in Section 2. Suppose that the dynamics and cost function in time $t \in [T]$ are given by

$$x_{t+1} = A_t x_t + B_t u_t + w_t \tag{8}$$

and

$$c_t(x_t, u_t) = \frac{1}{2} \left( (x_t)^\top Q_t x_t + (u_t)^\top R_t u_t \right). \tag{9}$$

Here, $(w_t : t \in [T])$ is a sequence of bounded and oblivious disturbances that is unknown to the online controller. [2] At each time step, the controller observes $(A_t, B_t, Q_t, R_t)$ for future $k$ time steps but all future disturbances are unknown. Since we assume $c_T \equiv 0$, the optimal $u_{T-1}$ is always 0 and the online control problem in episode $t$ actually terminates after the state $x_{T-1}$ is revealed.

We show how to apply Model Predictive Control (MPC) with robust predictions as the robust baseline in our framework [17, 62]. To define MPC with robust predictions, we first need to define the finite-time optimal control problem (FTOCP) solved by MPC at every step: For $t, t' \in [T]$, we define

$$\psi_{t,t'} \left( x_t, \left( w_{\tau|t} : \tau \in [t : t' - 1] \right); P_{t'} \right) = \underset{u_{t:(t'-1)|t}}{\arg\min} \sum_{\tau=t}^{t'-1} c_\tau \left( x_{\tau|t}, u_{\tau|t} \right) + \frac{1}{2} x_{t'|t}^\top P_{t'} x_{t'|t}$$

$$\text{s.t. } x_{\tau+1|t} = A_t x_{\tau|t} + B_t u_{\tau|t} + w_{\tau|t}, \forall \tau \in [t : t' - 1];$$

$$x_{t|t} = x_t,$$

Here, $\left( w_{\tau|t} : \tau \in [t : t' - 1] \right)$ can be viewed as the predicted future disturbances, and MPC with robust predictions sets them to zero vectors, therefore becomes robust against (potentially) adversarial environments with large disturbances $(w_t : t \in [nt])$. The term $x_{t'|t}^\top P_{t'} x_{t'|t}/2$ is a terminal cost that regularizes the last predictive state. To simplify the notation, we use the shorthand $\psi_{t,t'}(x_t; P) \coloneqq \psi_{t,t'}(x_t, 0_{\times(t'-t)}; P)$, where $0_{\times(t'-t)}$ denotes a sequence of $(t' - t)$ zeros, and

$$\psi_{t,t'}(x_t) \coloneqq \begin{cases} \psi_{t,t'}(x_t; P_{t'}) & \text{if } t' < T - 1, \\ \psi_{t,t'}(x_t; Q_{t'}) & \text{if } t' = T - 1. \end{cases}$$

With this notation, we define MPC with robust predictions formally in Algorithm 3. At each time step, it solves a $k$-step predictive FTOCP and commits the first control action in the optimal solution. Since the future disturbances are unknown, MPC predicts them to zero vectors. The terminal cost matrices $P_t$ are pre-determined.

We make some standard assumptions, following those in the literature of online control [35, 63, 19]. The first assumption is that the cost functions are well-conditioned and the dynamical matrices are uniformly bounded.

**Assumption 4.** *For any $t \in [T]$, we have $\|A_t\| \le a, \|B_t\| \le b, \|w_t\| \le d$, and*

$$\mu I_n \preceq Q_t \preceq \ell I_n, \mu I_m \preceq R_t \preceq \ell I_m, \mu I_n \preceq P \preceq \ell I_n.$$

---

[2]By oblivious, we mean the sequence $(w_t : t \in [T])$ is determined by the environment before the game starts and are not random.

**Algorithm 3** MPC with Robust Predictions (MPC$_k$)

**Initialize :** Prediction horizon $k$ and terminal cost matrix $P$.

7 **for** $t = 0, \ldots, T - 1$ **do**
8      Set $t' \leftarrow \min\{t + k, T - 1\}$
9      Observe $x_t$ and $(A_\tau, B_\tau, Q_\tau, R_\tau : \tau \in [t : t' - 1])$
10      Set action $u_t = \psi_{t,t'}(x_t)[u_{t|t}]$
11 **end**

The second assumption guarantees that for arbitrary bounded disturbance sequences $(w_t : t \in [T])$, there exists a controller that can stabilize the system.

**Assumption 5.** *For any $t, t' \in [T], t \leq t'$, define a block matrix*

$$
\Xi^t_{t,t'} := \begin{bmatrix} I & & & & \\ -A_t & -B_t & I & & \\ & & \ddots & & \\ & & & -A_{t'-1} & -B_{t'-1} & I \end{bmatrix}.
$$

*We assume $\sigma_{\min}(\Xi_{t,t'}) \geq \sigma$ for some positive constant $\sigma$, where $\sigma_{\min}(\cdot)$ denotes the smallest singular value of a matrix.*

The interpretation of Assumption 5 is as follows. It holds with $\sigma$ if and only if for any sequence $x_t, w_{t:t'-1}$ that satisfies $\left\| \left( (x_t)^\top, (w_t)^\top, \ldots, (w_{t'-1})^\top \right) \right\| \leq 1$, there exists a feasible trajectory $x_t, u_t, x_{t+1}, \ldots, u_{t'-1}, x_{t'}$ subject to

$$
\left\| \left( (x_t)^\top, (u_t)^\top, (x_{t+1})^\top, \ldots, (u_{t'-1})^\top, (x_{t'})^\top \right) \right\| \leq \frac{1}{\sigma}.
$$

Thus, Assumption 5 holds provided that there is an exponentially stabilizing controller. With these assumptions, we are ready to present our main result about MPC$_k$ in Theorem A.1.

**Theorem A.1.** *Suppose Assumptions 4 and 5 hold. Consider the case when the robust baseline policy $\overline{\pi}$ is MPC$_k$ (Algorithm 3), and for some $\overline{\lambda} \in (\lambda, 1)$, the prediction horizon $k$ satisfies that*

$$
k \geq \min \left\{ T, \frac{1}{2} \log \left( C^3 b a \lambda / (\overline{\lambda} - \lambda) \right) / \log(1/\lambda) \right\}.
$$

*We also assume that $x_0 = 0$, and $R_t \leq \overline{R}$. Then, the following holds for the robust baseline $\overline{\pi}$:*

    *(i) The Wasserstein robustness (Definition 3) holds globally with $s(t) = C(1 + C)(a + b)\overline{\lambda}^{t-1}$.*

    *(ii) The PROP controller (Algorithm 1) is always stable in the sense that*

$$
\|x_t\| \leq \overline{R}_x := \frac{C(d + b\overline{R})}{1 - \overline{\lambda}} \text{ and } \|u_t\| \leq \overline{R}_u := C\overline{R}_x + \overline{R}.
$$

    *(iii) The competitive ratio of the robust baseline MPC$_k$ satisfies that*

$$
\mathsf{ROB} \leq \frac{2\ell C^2 (1 + C^2)(1 + a^2 + b^2)}{\mu (1 - \overline{\lambda})^2}.
$$

*Here, the coefficients $C$ and $\lambda$ are given by $\lambda = \left( \frac{\overline{\sigma} - \underline{\sigma}}{\overline{\sigma} + \underline{\sigma}} \right)^{\frac{1}{2}}, C = \frac{4(\ell + 1 + a + b)}{\underline{\sigma}^2 \cdot \lambda}$, where*

$$
\underline{\sigma} := \min(\mu, 1) \cdot (a + b + 1) \cdot \sqrt{\frac{\ell}{2\mu\ell + \mu\sigma^2}}, \text{ and } \overline{\sigma} := \sqrt{2}(\ell + a + b + 1).
$$

The first result of Theorem A.1 shows that MPC$_k$ satisfies the Wasserstein robustness (see Definition 3), which is the critical assumption we require for any robust baseline policy. The second result guarantees that PROP (with MPC$_k$ as the robust baseline) will always stay in a bounded ball in the

Euclidean space as long as the radius $R_t$ is uniformly upper bounded. Thus, we can assume $\mathcal{X}$ and $\mathcal{U}$ are compact without loss of generality. The third result gives an upper bound of the robust competitive ratio ROB, which in this application is a special deterministic case of the considered ratio of expectation (ROE) in the general results. With the settings above, we conclude that in the grey-box setting, PROP with GREY-BOX Procedure can be 1-consistent and $\left( \frac{2\ell C^2(1+C^2)(1+a^2+b^2)}{\mu(1-\overline{\lambda})^2} + o(1) \right)$-robust. We defer the detailed proof of Theorem A.1 to Appendix E.

## A.2 Baseline Policies for MDPs with Finite State/Action Spaces

Our second example focuses on an MDP environment $(\mathcal{S}, \mathcal{A}, (\mathbb{P}_t : t \in [T]), (c_t : t \in [T]), T)$ with a finite state space $\mathcal{S}$ and a finite action space $\mathcal{A}$. Given a policy $\overline{\pi}_t : \mathcal{S} \to \Delta(\mathcal{A})$ for $t \in [T]$, let $(\overline{\mathbb{P}}_t)$ denote the state transition probability that maps $\mathcal{S}$ to $\Delta(\mathcal{S})$, which is defined as

$$\overline{\mathbb{P}}_t(s; s') = \sum_{a \in \mathcal{A}} \overline{\pi}_t(s; a) \mathbb{P}_t(s, a; s').$$

We consider the setting when every entry of $\overline{\mathbb{P}}_t$ is strictly positive. Under this assumption, one can show that the one-step transition probability $\overline{\mathbb{P}}_t$ is a contractive mapping in total variance distance [64]. We state this result formally in Lemma 1. To simplify the notation, for any $0 \le t \le t' < T$, we define the multi-step transition matrix as $\overline{\mathbb{P}}_{t:t'} := \overline{\mathbb{P}}_t \overline{\mathbb{P}}_{t+1} \cdots \overline{\mathbb{P}}_{t'}$.

**Lemma 1.** *Under the assumption that* $\min_{t \in [T]} \min_{s,s' \in \mathcal{S}} \overline{\mathbb{P}}_t(s; s') \ge \epsilon$*, for any* $0 \le t \le t' < T$ *and distributions* $\mu, \nu \in \Delta(\mathcal{S})$*, we have that*

$$\mathsf{TV} \left( \mu^\top \overline{\mathbb{P}}_{t:t'}, \nu^\top \overline{\mathbb{P}}_{t:t'} \right) \le \lambda^{t'-t} \mathsf{TV}(\mu, \nu), \tag{10}$$

*where* $\lambda = 1 - |\mathcal{S}| \epsilon$.

Lemma 1 follows from Proposition 5 in [65]. In the case that not every entry of $\overline{\mathbb{P}}_t$ is strictly positive, but the entries of $\overline{\mathbb{P}}_{t:t+d}$ are strictly positive for some constant $d \in \mathbb{Z}_+$, we can still obtain a similar contraction property as Lemma 1 with a weaker decay rate $\lambda$. Note that the exponential contractive property in Lemma 1 is different with the one in Wasserstein robustness (Definition 3) because the distance between distributions are measured by total variance instead of Wasserstein distance. To convert it into the form required by Wasserstein robustness, we need to define an underlying metric for the discrete state/action space.

Without loss of generality, we assume $\mathcal{X} := \{e_i : i = 1, \ldots, |S|\} \subseteq \mathbb{R}^{|\mathcal{S}|}$, where each element of $\mathcal{X}$ corresponds to a unique state in $\mathcal{S}$. Here, each $e_i$ is an indicator vector of $\mathbb{R}^{|\mathcal{S}|}$ defined as

$$e_i(j) = \begin{cases} 1 & \text{if } i = j, \\ 0 & \text{otherwise.} \end{cases}$$

Since a policy in the discrete MDP maps $\mathcal{S}$ to $\Delta(\mathcal{A})$, we set $\mathcal{U} = \Delta(\mathcal{A}) \subseteq \mathbb{R}^{|\mathcal{A}|}$, which denotes the distribution of actions and is compact and convex. To define the Wasserstain distance, we adopt $\ell_1$ distance as the metric on the state space $\mathcal{X}$, action space $\mathcal{U}$, and state-action space $\mathcal{X} \times \mathcal{U}$, i.e.,

$$\|(x, u) - (x', u')\|_1 = \|x - x'\|_1 + \|u - u'\|_1, \text{ for all } x, x' \in \mathcal{X}, u, u' \in \mathcal{U}.$$

Using these definitions, we can use the contraction property in the TV distance (Lemma 1) to establish the Wasserstein robustness of the baseline policy $\overline{\pi}$.

**Theorem A.2.** *Suppose the Markov chain on state space* $\mathcal{S}$ *induced by the baseline policy* $\overline{\pi} = (\overline{\pi}_t : t \in [T])$ *satisfies that* $\overline{\mathbb{P}}_t(s; s') \ge \epsilon$ *for all* $t \in [T]$ *and* $s, s' \in \mathcal{S}$*, then Wasserstein robustness (Definition 3) holds globally with* $s(t) = 2\lambda^{t-1}$*, where* $\lambda = 1 - |\mathcal{S}| \epsilon$.

We defer the proof of Theorem A.2 to Appendix F. Theorem A.2 shows that the Wasserstein robustness in Definition 3 is general enough to capture a wide class of baseline policies in finite state/action settings. It also enables comparison between our results and previous studies that assume discrete state/action spaces [66, 9, 7].

## A.3 Numerical Results

In light of the applications detailed in Appendix A.1, we present two case studies. We consider linear dynamics as a specific instance of an MDP and use the MPC described in Algorithm 3 as our robust baseline.

### A.3.1 Basic Settings

**Dynamics.** We investigate the impact of the hyper-parameter $\beta$ in the robustness budget $R_t$ in (7) by considering the following update rule:

$$\begin{bmatrix} d_{t+1} \\ v_{t+1} \end{bmatrix} = A \begin{bmatrix} d_t \\ v_t \end{bmatrix} + Bu_t + w_t, \tag{11}$$

which is cast in the canonical form (8). The system matrices $A$ and $B$ are defined as

$$A := \begin{bmatrix} 1 & 0 & 0.2 & 0 \\ 0 & 1 & 0 & 0.2 \\ 0 & 0 & 1 & 0 \\ 0 & 0 & 0 & 1 \end{bmatrix}, \quad B := \begin{bmatrix} 0 & 0 \\ 0 & 0 \\ 0.2 & 0 \\ 0 & 0.2 \end{bmatrix}, \tag{12}$$

and $w_t := Ay_t - y_{t+1}$, where $(y_t : t \in [T])$ specifies an unknown trajectory to be tracked. The choice of $A, B$ and $(w_t : t \in [T])$ specifies a two-dimensional robot tracking problem as detailed in [67, 17]. In this application, the robot controller maneuvers along a fixed but unknown trajectory, given by $(y_t : t \in [T])$. At each time $t \in [T]$, the robot controller needs to decide an acceleration action $u_t$, without knowing the desired location $y_t$. It can only access the past trajectories $(y_\tau : \tau \in [t])$. The location of the robot controller at time $t + 1$, denoted $l_{t+1} \in \mathbb{R}^2$, is determined by its prior location and its velocity $v_t \in \mathbb{R}^2$ according to $l_{t+1} = l_t + 0.2v_t$. Furthermore, at each subsequent time $t$, the controller has the ability to apply an adjustment $u_t$ to alter its velocity, resulting in $v_{t+1} = v_t + 0.2u_t$ at the next time step. This system can be reformulated as (11) by letting $x_t = l_t - y_t$, the tracking error between the current location at time $t$ and the desired location $y_t$.

To efficiently track the trajectory, we use quadratic costs as in (9) with

$$Q := \begin{bmatrix} 1 & 0 & 0 & 0 \\ 0 & 1 & 0 & 0 \\ 0 & 0 & 0 & 0 \\ 0 & 0 & 0 & 0 \end{bmatrix}, \quad R := \begin{bmatrix} 10^{-2} & 0 \\ 0 & 10^{-2} \end{bmatrix}. \tag{13}$$

With the settings above, we encapsulate a Gym environment [68] with action space and state space defined as hyper-cubes in $\mathbb{R}^2$ and $\mathbb{R}^4$ such that each action/state coordinate is within $[-100, 100]$.

**MPC Baseline $\bar{\pi}$.** With predictions $(\widetilde{w}_t : t \in [T])$ of the perturbations satisfy $\widetilde{w}_t = 0$ for all $t \in [T]$, with a terminal cost matrix $P$ as the solution of the discrete algebraic Riccati equation (DARE), the MPC baseline in Algorithm 3 can be stated as the following linear quadratic regulator $\bar{\pi}_{\mathsf{MPC}}(x_t) = -Kx_t$ where $K := (R + B^\top PB)^{-1}B^\top PA$.

**Machine-Learned Policy $\widetilde{\pi}$.** We use the deep deterministic policy gradient (DDPG) algorithm [69] to generate machine learned advice and Q-value functions, with hyper-parameters set as in Table 2.

Table 2: Hyper-parameters used in DDPG.

| Parameter | Value |
|---|---|
| Maximal number of episodes | $10^3$ |
| Episode length | $10^2$ |
| Discount factor | 1.0 |
| Actor network learning rate | $10^{-3}$ |
| Critic network learning rate | $10^{-3}$ |
| Soft target update parameter | $10^{-3}$ |
| Replay buffer size | $10^6$ |
| Minibatch size | 128 |

**PROP Implementation** In our empirical implementation of PROP, we set $L_Q = 1$ in (6) and use $|\delta_t|$ at each $t$-th time step, instead of $\sum_{s=1}^t \delta_s$ to generate more stable results.

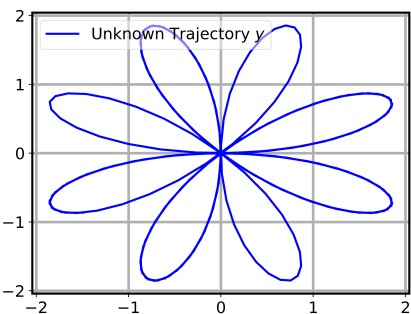

Figure 3: Unknown trajectory $y$ in the case study that illustrates the impact of $\beta$.

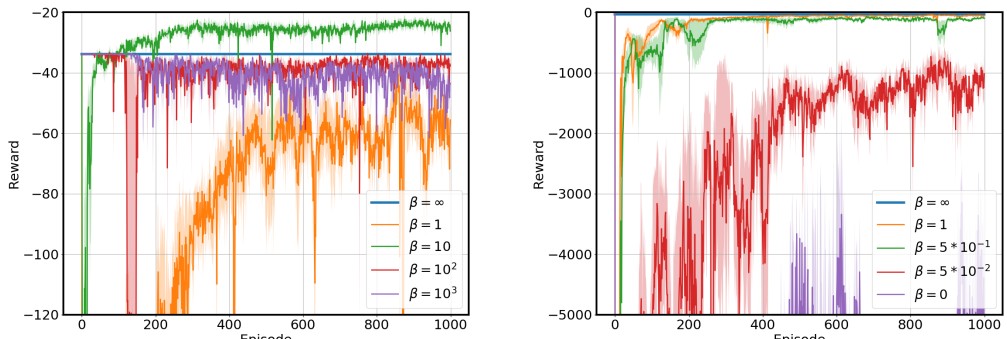

Figure 4: Average awards with varying choices of the hyper-parameter $\beta$ in the robustness budget of PROP. Shadow area depicts the range of standard deviations for $5$ random tests. Left: $\beta = 1, 10, 10^2, 10^3$, and $\infty$ (directly applying the MPC baseline); Right: $\beta = 0, 0.05, 0.5, 1$, and $\infty$.

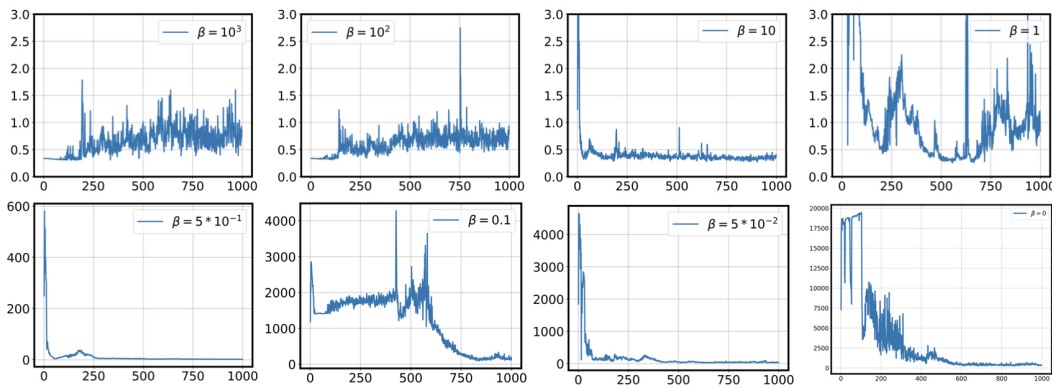

Figure 6: Average approximate TD-error (see (6)) with different choices of the hyper-parameter $\beta$ in the robustness budget of PROP. Shadow area depicts the range of standard deviations for $5$ random tests.

### A.3.2 Case Studies

**Impact of Hyper-Parameter $\beta$.** With the basic settings described above, we delve into the effects of the hyper-parameter $\beta$ on the selection of the robustness budget as in (7). This, in turn, influences the average rewards, projection radii, and the approximate TD-error for PROP.

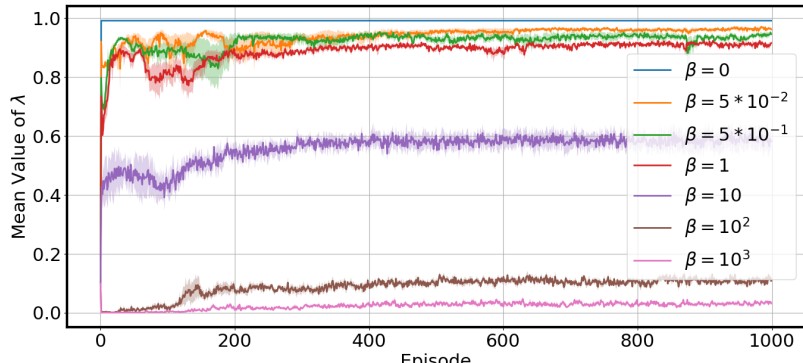

Figure 5: The influence of hyper-parameter $\beta$ on the projection radii $(R_t : t \in [T])$. The shaded region represents the standard deviation range from 5 random tests. As $\beta$ increases, the average trust coefficient $\lambda(R_t)$ decreases.

We set the unknown $(y_t : t \in [T])$ to be tracked as a rose-shaped trajectory shown in Figure 3:

$$y_t := \begin{bmatrix} 2\cos\left(\frac{t}{20}\right)\sin\left(\frac{t}{5}\right) \\ 2\sin\left(\frac{t}{20}\right)\sin\left(\frac{t}{5}\right) \end{bmatrix}, \quad t \in [T]. \tag{14}$$

We vary the value of $\beta$ from 0 up to $\infty$. It is important to highlight that when $\beta = 0$, PROP operates the same as the pure DDPG in our experiments. In contrast, when $\beta = \infty$, PROP is equivalent to the MPC baseline discussed earlier. Arbitrary exploration in the action space will lead to unstable states, causing the pure DDPG to remain non-convergent throughout its training process. From our experiments, we observe that setting $\beta$ between 5 and 25 yields the largest average reward. The results are summarized in Figure 4.As noted in the proof of Lemma 3 in Appendix B.2, the action given by PROP at each time $t \in [T]$ can be written as

$$u_t = \lambda\left(R_t\right)\widetilde{\pi}_t\left(x_t\right) + \left(1 - \lambda\left(R_t\right)\right)\overline{\pi}_t\left(x_t\right)$$

where $\lambda\left(R_t\right) := \min\left\{1, R_t/\|\widetilde{\pi}_t\left(x_t\right) - \overline{\pi}_t\left(x_t\right)\|_{\mathcal{U}}\right\}$ serves as a *trust coefficient* between 0 and 1. Here, $R_t$ is the robustness budget defined in (7). In Figure 5 we illustrate the behavior of $\lambda(R_t)$ averaged over all time steps and tests. Likewise, Figure 6 displays the evolution of the approximate TD-error with various selections of the hyper-parameter $\beta$, averaged over all time steps and tests. Notably, a distinct convergence of the approximate TD-error is evident when $\beta = 10$, which also yields high average rewards, shown in Figure 4. It's worth noting, however, that we did not actively optimize for $\beta$.

**Non-Stationary Environment**    In a subsequent experiment, we address scenarios where there is a distribution shift in the underlying MDP. We use the same matrices $A, B, Q,$ and $R$ in (12) and (13).

For each $w_t$ in (11), we treat it as an independent Gaussian vector. Specifically, every entry $w_t(i)$ of $w_t$ is considered as an independent Gaussian random variable. For the first 700 episodes, each $w_t(i)$ is sampled from a normal distribution $\mathcal{N}(\mu, \sigma)$ where $\mu = 0.5$ and $\sigma = 0.05$. However, in the last 300 episodes, we adjust $\mu$ to $-0.5$.

In the context of this nonstationary MDP, Figure 7 illustrates the reward recovery after the occurrence of a distribution shift for varying choices of $\beta$. The two top figures highlight the average rewards: the top-left figure corresponds to an action space of $\mathcal{U} = [-100, 100]$, while the top-right is set to $\mathcal{U} = [-5, 5]$. On the bottom, the left figure presents the average behavior of $\lambda(R_t)$, and the right one illustrates the average approximate TD-error for the case $\mathcal{U} = [-100, 100]$ With $\beta = 10$ for $\mathcal{U} = [-100, 100]$ and $\beta = 1$ for $\mathcal{U} = [-5, 5]$ respectively, there is an evident near-optimal tradeoff between consistency and robustness. PROP consistently achieves notable average rewards before the distribution shift and showcases a swift recovery afterwards, validating the algorithm's efficacy. Similar to the fist set of experiments, it is worth mentioning that we did not explicitly fine-tune the value of $\beta$.

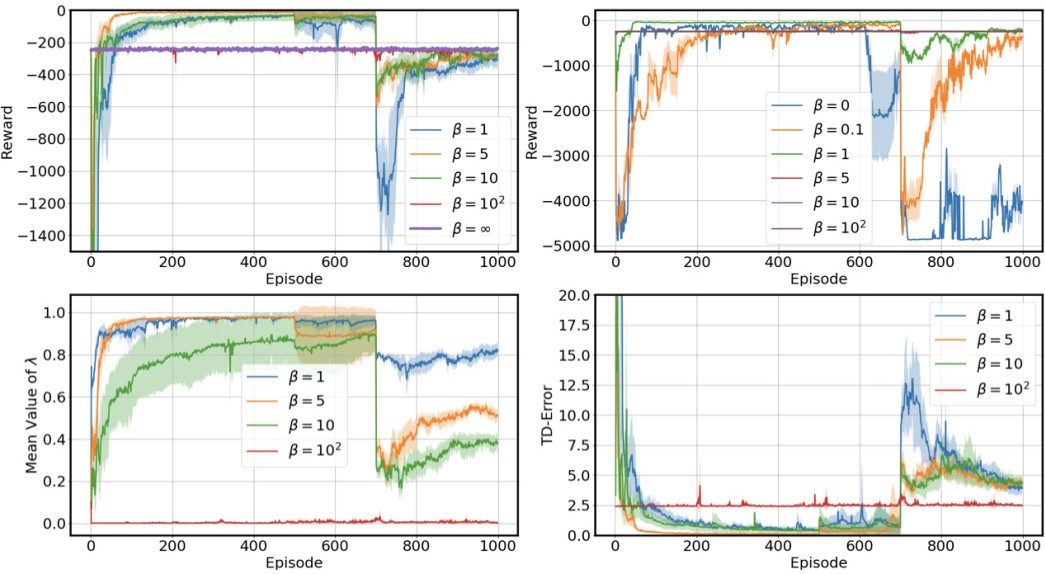

Figure 7: Stability against distribution shifts for different settings of the hyper-parameter $\beta$.

# B    Useful Lemmas

In this appendix, we present results that will be used when proving our main theorems.

## B.1    Perturbation Lemma

We first prove the following perturbation lemma as a robustness guarantee, which holds for both the black-box (Section 4.1) and grey-box (Section 4.2) settings.

**Lemma 2** (Perturbation Lemma). *Under Assumption 1 and 2, the dynamic regret of Algorithm* (1) *(denoted by* PROP*) can be bounded by* $\mathsf{DR}(\mathsf{PROP}) \leq \mathcal{O}\left((\mathsf{ROB}-1)T\right) + L_{\mathrm{C}} C_s \sum_{t \in [T]} \mathbb{E}\left[(R_t)^p\right]^{1/p}$ *for constants* $L_{\mathrm{C}}, C_s > 0$.

*Proof.* Our proof consists of two parts. We first bound the Wasserstein distance between joint action-state distributions for the robust baseline and PROP. Next, we bound the dynamic regret.

**Step 1.  Wasserstein Distance between Joint Action-State Distributions.** Denote by $\pi$ and $\overline{\pi}$ the PROP policy (Algorithm 1) and an $r$-locally $p$-Wasserstein-robust policy respectively. For any step $t, t' \in [T]$ with $t' \leq t$, denote by $\overline{\rho}_{t|t'}$ the state-action distribution generated by applying the actions given by Algorithm 1 until step $t'$ and applying the actions generated by the $r$-locally $p$-Wasserstein-robust policy afterwards until step $t$. Let $\rho_t$ and $\overline{\rho}_t$ be the state-action distributions corresponding to Algorithm 1 and the $r$-locally $p$-Wasserstein robust baseline at each step $t \in [T]$. Using the triangle inequality, the state-action distribution difference between $\rho_t$ and $\overline{\rho}_t$ in terms of the Wasserstein $p$-distance satisfies:

$$W\left(\rho_t, \overline{\rho}_t\right) \leq \sum_{\tau=0}^{t-1} W\left(\overline{\rho}_{t|t-\tau}, \overline{\rho}_{t|t-\tau-1}\right) \tag{15}$$

with $\overline{\rho}_{t|0} := \overline{\rho}_t$ and $\overline{\rho}_{t|t} := \rho_t$. Abuse the notation $\pi$ and denote by $\overline{\pi}_{t-\tau:t}$ an operator on state-action distributions for applying the control baselines $\overline{\pi}_{t-\tau}, \overline{\pi}_{t-\tau+1}, \ldots, \overline{\pi}_t$ consecutively. Continuing from (15), it follows that for any $t \in [T]$ and $0 \leq \tau < t$,

$$W\left(\overline{\rho}_{t|t-\tau}, \overline{\rho}_{t|t-\tau-1}\right) = W\left(\overline{\pi}_{t-\tau:t}\left(\overline{\rho}_{t-\tau|t-\tau}\right), \overline{\pi}_{t-\tau:t}\left(\overline{\rho}_{t-\tau|t-\tau-1}\right)\right)$$

$$\leq s(\tau) W\left(\overline{\rho}_{t-\tau|t-\tau}, \overline{\rho}_{t-\tau|t-\tau-1}\right) \tag{16}$$

where $\overline{\rho}_{t-\tau|t-\tau} := \rho_{t-\tau}$ and in (16) we have used the assumption of the $r$-locally $p$-Wasserstein-robust policy (Definition 3). The assumption can be applied because by definition, for all $t \in [T]$, the Wasserstein $p$-distance between $\overline{\rho}_{t-\tau|t-\tau}$ and $\overline{\rho}_{t-\tau|t-\tau-1}$ can be bounded by

$$W\left(\overline{\rho}_{t-\tau|t-\tau}, \overline{\rho}_{t-\tau|t-\tau-1}\right)$$

$$\leq \mathbb{E}\left[\|(x_{t-\tau-1}, \pi_{t-\tau}(x_{t-\tau-1})) - (x_{t-\tau-1}, \overline{\pi}_{t-\tau}(x_{t-\tau-1}))\|^p_{\mathcal{X} \times \mathcal{U}}\right]^{1/p} \tag{17}$$

$$= \mathbb{E}\left[\|\pi_{t-\tau}(x_{t-\tau-1}) - \overline{\pi}_{t-\tau}(x_{t-\tau-1})\|^p_{\mathcal{U}}\right]^{1/p} \tag{18}$$

$$\leq \mathbb{E}\left[(R_{t-\tau})^p\right]^{1/p} \leq r, \tag{19}$$

where in (17) we have used the definition

$$W_p(\mu, \nu) := \left(\inf_{J \in \mathcal{J}(\mu,\nu)} \int \|(x,u) - (x',u')\|^p_{\mathcal{X} \times \mathcal{U}} \mathrm{d}J\left((x,u),(x',u')\right)\right)^{1/p}.$$

Since $\|(x,u) - (x',u')\|_{\mathcal{X} \times \mathcal{U}} := \|x - x'\|_{\mathcal{X}} + \|u - u'\|_{\mathcal{U}}$, (18) follows. Finally, we obtain (19) considering the projection constraint in Algorithm 1. Combining (19) with (15),

$$W_p\left(\rho_t, \overline{\rho}_t\right) \leq \sum_{\tau=0}^{t-1} s(\tau) \mathbb{E}_{P,\pi}\left[(R_{t-\tau})^p\right]^{1/p}. \tag{20}$$

**Step 2. Dynamic Regret Analysis.** Since the cost functions $(c_t : t \in [T])$ are Lipschitz continuous with a Lipschitz constant $L_C$, using the Kantorovich-Rubinstein duality theorem [70], since $W_p(\mu, \nu) \leq W_q(\mu, \nu)$ for all $1 \leq p \leq q < \infty$, for all $t \in [T]$,

$$\mathbb{E}_{(x,u)\sim\rho_t}\left[c_t(x,u)\right] - \mathbb{E}_{(x,u)\sim\overline{\rho}_t}\left[c_t(x,u)\right]$$

$$\leq \sup_{\|f\|_L \leq L_C} \mathbb{E}_{(x,u)\sim\rho_t}\left[f(x,u)\right] - \mathbb{E}_{(x,u)\sim\overline{\rho}_t}\left[f(x,u)\right] \leq L_C W_p\left(\rho_t, \overline{\rho}_t\right), \tag{21}$$

where $\|\cdot\|_L$ denotes the Lipschitz semi-norm and the supremum is over all Lipschitz continuous functions $f$ with a Lipschitz constant $L_C$. Therefore, the difference between the expected cost of Algorithm 1, denoted by $\pi$, and the baseline policy $\overline{\pi}$ satisfies

$$J(\pi) - J(\overline{\pi}) = \sum_{t \in [T]} \mathbb{E}_{(x,u)\sim\rho_t}\left[c_t(x,u)\right] - \mathbb{E}_{(x,x)\sim\overline{\rho}_t}\left[c_t(x,u)\right]$$

$$\leq L_C \sum_{t \in [T]} \sum_{\tau=0}^{t-1} s(\tau) \mathbb{E}_{P,\pi}\left[(R_{t-\tau})^p\right]^{1/p}$$

$$\leq L_C C_s \sum_{t \in [T]} \mathbb{E}_{P,\pi}\left[(R_t)^p\right]^{1/p} \tag{22}$$

where we have used the assumption of the $r$-locally robustness policy so that $\sum_{t \in [T]} s(t) \leq C_s$ for some constant $C_s > 0$. Moreover, since the robust baseline $\overline{\pi}$ has a ratio of expectations bound such that $\frac{J(\overline{\pi})}{J^\star} \leq \mathsf{ROB}$. Using Assumption 1, from (22), we obtain

$$\mathsf{DR}(\mathsf{PROP}) := J(\pi) - J^\star \leq \mathcal{O}\left((\mathsf{ROB} - 1)T\right) + L_C C_s \sum_{t \in [T]} \mathbb{E}_{P,\pi}\left[(R_t)^p\right]^{1/p}.$$

$\square$

## B.2 Projection Lemma

The following lemma implies a useful consistency bound. It is worth noting that the lemma also holds if PROP adopts an alternative approach instead of projecting the actions as shown in (5):

$$u_t \in \arg\min_{v \in \mathsf{U}} \widetilde{Q}_t(x_t, v) \text{ subject to } \|\overline{u}_t - v\| \leq R_t$$

Implementing the projection rule in PROP can significantly reduce computational complexity, particularly when dealing with non-convex Q-advice.

**Lemma 3** (Projection Lemma). *Under Assumption 3, the actions and states $(x_t, u_t)$ at $t \in [T]$ and $t \in [T]$ generated by* PROP *(Algorithm 1) satisfy*

$$Q_t^\star (x_t, u_t) - \inf_{v \in \mathcal{U}} Q_t^\star (x_t, v) \leq L_Q \left( [\eta_t (x_t) - R_t]^+ \right) + \mu_t (x_t, u_t) \tag{23}$$

*where $\eta_t (x) := \|\widetilde{\pi}_t (x) - \overline{\pi}_t (x)\|_{\mathcal{U}}$, and $Q^\star$ denotes the optimal Q-value functions satisfying the Bellman optimality equations in* (3).

*Proof.* Let $\left( \widetilde{Q}_t : t \in [T], t \in [T] \right)$ be the Q-value advice used in Algorithm 1, denoted by $\pi$. Since $\zeta_t^Q := \widetilde{Q}_t(x_t, u_t) - Q_t^\star(x_t, u_t)$, we have, for any $t \in [T]$ and $(x_t, u_t)$ generated by Algorithm 1,

$$Q_t^\star (x_t, u_t) - \inf_{v \in \mathcal{U}} Q_t^\star (x_t, v) = \widetilde{Q}_t (x_t, u_t) - Q_t^\star (x_t, u_t^\star) - \zeta_t^Q. \tag{24}$$

Let $\widetilde{\pi}_t (x_t) := \inf_{u \in \mathcal{U}} \widetilde{Q}_t (x_t, u)$. Note that since $\mathcal{U}$ is convex and compact, the projection step in the PROP policy is equivalent to (we choose the $u_t$ on the line formed by $\widetilde{\pi}_t (x_t)$ and $\overline{\pi}_t (x_t)$ if there are ties in the projection solution)

$$u_t = \lambda (R_t) \widetilde{\pi}_t (x_t) + (1 - \lambda (R_t)) \overline{\pi}_t (x_t),$$

where $\lambda (R_t) := \min \{1, R_t / \|\widetilde{\pi}_t (x_t) - \overline{\pi}_t (x_t)\|_{\mathcal{U}}\}$. Write $\eta_t (\cdot) := \|\widetilde{\pi}_t (\cdot) - \overline{\pi}_t (\cdot)\|_{\mathcal{U}}$. Therefore, the Q-value advice satisfies

$$\begin{aligned}
\widetilde{Q}_t (x_t, u_t) &\leq \widetilde{Q}_t (x_t, \widetilde{\pi}_t (x_t)) + L_Q \|\widetilde{\pi}_t (x_t) - u_t\| \\
&\leq \widetilde{Q}_t (x_t, \widetilde{\pi}_t (x_t)) + L_Q (1 - \lambda (R_t)) \eta_t (x_t) \\
&= \widetilde{Q}_t (x_t, \widetilde{\pi}_t (x_t)) + L_Q \left( 1 - \min \left\{ 1, \frac{R_t}{\eta_t (x_t)} \right\} \right) \eta_t (x_t) \\
&\leq \widetilde{Q}_t (x_t, \widetilde{\pi}_t (x_t)) + L_Q (\eta_t (x_t) - R_t),
\end{aligned} \tag{25}$$

where (25) follows since by construction $R_t \leq \eta_t$ for all $t \in [T]$. Let $\widetilde{u}_t$ denote $\widetilde{\pi}_t (x_t)$. Continuing from (24),

$$\begin{aligned}
&Q_t^\star (x_t, u_t) - \inf_{v \in \mathcal{U}} Q_t^\star (x_t, v) \\
&= \widetilde{Q}_t (x_t, u_t) - Q_t^\star (x_t, u_t^\star) - \zeta_t^Q \\
&\leq \widetilde{Q}_t (x_t, \widetilde{u}_t) - Q_t^\star (x_t, u_t^\star) + L_Q [\eta_t (x_t) - R_t]^+ - \zeta_t^Q \tag{26} \\
&\leq L_Q [\eta_t (x_t) - R_t]^+ + \mu_t \tag{27}
\end{aligned}$$

where (26) follows from (25). Since $\zeta_t^V := \widetilde{Q}_t(x_t, \widetilde{u}_t) - Q_t^\star(x_t, u_t^\star)$, $\widetilde{u}_t$ minimizes $\widetilde{Q}_t$, and $\mu_t := \zeta_t^V - \zeta_t^Q$, we obtain (27). $\qquad\square$

### B.3 Analysis of Approximate TD-Error

The following result that rewrites the approximate TD-error (c.f. (6)) is useful.

**Lemma 4.** *Consider the approximate TD-error in* (6) *such that*

$$\delta_t (u_{t-1}, x_{t-1}, x_t) := c_{t-1} (x_{t-1}, u_{t-1}) + \inf_{v \in \mathcal{U}} \widetilde{Q}_t (x_t, v) - \widetilde{Q}_{t-1} (x_{t-1}, u_{t-1}).$$

*It follows that for any $t \in [T]$,*

$$\mathbb{E}_{P,\pi} [\delta_t (u_{t-1}, x_{t-1}, x_t)] = \mathbb{E}_{P,\pi} \left[ \zeta_t^V (x_t) - \zeta_{t-1}^Q (x_{t-1}, u_{t-1}) \right],$$

*where $\zeta_{-1}^Q = 0$, $\zeta_t^Q$ and $\zeta_t^V$ are defined as*

$$\begin{aligned}
\zeta_t^Q (x_t, u_t) &:= \widetilde{Q}_t (x_t, u_t) - Q_t^\star (x_t, u_t), \\
\zeta_t^V (x_t) &:= \inf_{v \in \mathcal{U}} \widetilde{Q}_t (x_t, v) - \inf_{v \in \mathcal{U}} Q_t^\star (x_t, v).
\end{aligned}$$

*Proof.* Taken expectation with randomness over the action and state trajectories, for any $t \in [T]$,

$$\mathbb{E}_{P,\pi}\left[\delta_t\left(u_{t-1}, x_{t-1}, x_t\right)\right]$$

$$:=\mathbb{E}_{P,\pi}\left[c_{t-1}\left(x_{t-1}, u_{t-1}\right)\right] + \mathbb{E}_{P,\pi}\left[\inf_{v\in\mathcal{U}} \widetilde{Q}_t\left(x_t, v\right)\right] - \mathbb{E}_{P,\pi}\left[\widetilde{Q}_{t-1}\left(x_{t-1}, u_{t-1}\right)\right]$$

$$=\mathbb{E}_{P,\pi}\left[\inf_{v\in\mathcal{U}} \widetilde{Q}_t(x_t, v) - \mathbb{P}_{t-1}V_t^{\star}(x_{t-1}, u_{t-1})\right] + \mathbb{E}_{P,\pi}\left[Q_{t-1}^{\star}(x_{t-1}, u_{t-1}) - \widetilde{Q}_{t-1}(x_{t-1}, u_{t-1})\right] \tag{28}$$

$$=\mathbb{E}_{P,\pi}\left[\inf_{v\in\mathcal{U}} \widetilde{Q}_t(x_t, v) - \mathbb{E}_{x'\sim P_t(\cdot|x_{t-1}, u_{t-1})}\inf_{v\in\mathcal{U}} Q_t^{\star}(x', v)\right]$$

$$+ \mathbb{E}_{P,\pi}\left[Q_{t-1}^{\star}(x_{t-1}, u_{t-1}) - \widetilde{Q}_{t-1}(x_{t-1}, u_{t-1})\right]$$

$$=\mathbb{E}_{P,\pi}\left[\zeta_t^V(x_t) - \zeta_{t-1}^Q(x_{t-1}, u_{t-1})\right]$$

where we have used the Bellman optimality equations (3) to derive (28).

$\square$

Next, we present our analysis of the black-box setting by proving Theorem 5.1 and Theorem 5.3.

## C  Black-Box Consistency and Robustness Analysis

### C.1  Proof of Theorem 5.1

Consider an MDP model with Assumption 1,2, and 3. We prove the theorem below.

**Theorem C.1.** *Suppose the machine-learned policy $\widetilde{\pi}$ is $(\infty, \varepsilon)$-consistent. The expected dynamic regret of* PROP *with the* BLACK-BOX *Procedure is bounded by*

$$\mathrm{DR}(\mathrm{PROP}) \leq \min\left\{\mathcal{O}(\varepsilon) + \mathcal{O}((1-\lambda)\gamma T), \mathcal{O}\left((\mathrm{ROB} + \lambda\gamma - 1)T\right)\right\}$$

*where $\varepsilon$ is defined in (4), $\gamma$ is the diameter of the action space $\mathcal{U}$, $T$ is the length of the time horizon, ROB is the ratio of expectations of the robust baseline $\bar{\pi}$, and $0 \leq \lambda \leq 1$ is a hyper-parameter.*

**Consistency Analysis.** To show the first bound in Theorem 5.1 regarding the consistency result, we consider the following steps. For any $t \in [T]$, denote by $(x_t, u_t)$ the corresponding state and action generated by the projection pursuit policy PROP, denoted by $\pi$. The Bellman optimality equations (3) imply:

$$Q_t^{\star}\left(x_t, u_t\right) = c_t\left(x_t, u_t\right) + \mathbb{E}_P\left[\inf_{v\in\mathcal{U}} Q_{t+1}^{\star}\left(x_{h+1}, v\right)\Big|x_t, u_t\right]. \tag{29}$$

Therefore the dynamic regret of the projection pursuit policy $\pi$ can be rewritten as

$$\mathrm{DR}(\mathrm{PROP}) = J(\pi) - J^{\star} = \left(\mathbb{E}_{P,\pi}\left[\sum_{t=0}^{T-1} c_t\left(x_t, u_t\right)\right] - \inf_{v\in\mathcal{U}} Q_{t,0}^{\star}\left(x_0, v\right)\right). \tag{30}$$

Combining (30) with (29), we obtain the following cost-difference bound:

$$\mathrm{DR}(\mathrm{PROP})$$

$$=\sum_{t=0}^{T-1}\left(\mathbb{E}_{P,\pi}\left[Q_t^{\star}\left(x_t, u_t\right)\right] - \mathbb{E}_P\left[\inf_{v\in\mathcal{U}} Q_{t+1}^{\star}\left(x_{t+1}, v\right)\right]\right) - \inf_{v\in\mathcal{U}} Q_{t,0}^{\star}\left(x_0, v\right)$$

$$=Q_{t,0}^{\star}\left(x_0, u_0\right) - \inf_{v\in\mathcal{U}} Q_{t,0}^{\star}\left(x_0, v\right) + \sum_{t=1}^{T-1}\left(\mathbb{E}_{P,\pi}\left[Q_t^{\star}\left(x_t, u_t\right)\right] - \mathbb{E}_P\left[\inf_{v\in\mathcal{U}} Q_t^{\star}\left(x_t, v\right)\right]\right)$$

$$=\sum_{t=0}^{T-1}\mathbb{E}_{P,\pi}\left[Q_t^{\star}\left(x_t, u_t\right) - \inf_{v\in\mathcal{U}} Q_t^{\star}\left(x_t, v\right)\right].$$

Recall that for the BLACK-BOX Procedure in Section 4.1, the robustness budget is set as $R_t = \lambda \eta_t$ for all $t \in [T]$. Applying the bound in Lemma 3 gives the following consistency bound:

$$\mathsf{DR}(\mathsf{PROP}) = \mathcal{O}\left( \sum_{t=0}^{T-1} \left( [\eta_t(x_t) - R_t]^+ \right) + \mathbb{E}_{P,\pi}[\mu_t] \right) \leq \mathcal{O}\left( \sum_{t=0}^{T-1} \mathbb{E}_{P,\pi}[\mu_t] \right) + \mathcal{O}\left( (1-\lambda)\gamma T \right)$$

since $\eta_t(x_t) \leq \gamma$ for all $t \in [T]$, and

$$\mu_t = \zeta_t^V - \zeta_t^Q = \widetilde{Q}_t(x_t, \widetilde{u}_t) - Q_t^\star(x_t, u_t^\star) - \left( \widetilde{Q}_t(x_t, u_t) - Q_t^\star(x_t, u_t) \right)$$

$$\leq \left\| \inf_{v \in \mathcal{U}} \widetilde{Q}_t(x_t, v) - \inf_{v \in \mathcal{U}} Q_t^\star(x_t, v) \right\|_\infty + \left\| \widetilde{Q}_t(x_t, u_t) - Q_t^\star(x_t, u_t) \right\|_\infty.$$

Noting that the machine-learned policy $\widetilde{\pi}$ is $(\infty, \varepsilon)$-consistent, we obtain $\sum_{t=0}^{T-1} \mathbb{E}_{P,\pi}[\mu_t] \leq \varepsilon$. Hence,

$$\mathsf{DR}(\mathsf{PROP}) = \mathcal{O}(\varepsilon) + \mathcal{O}\left( (1-\lambda)\gamma T \right). \tag{31}$$

**Robustness Analysis.** Note that for any $t \in [T]$, $\eta_t(x_t) \leq \gamma$, where $\gamma$ is the diameter of the compact action space $\mathcal{U}$. Hence, noting the black-box setting of the robustness budget $R_t = \lambda \eta_t$ for all $t \in [T]$ and applying Lemma 2, the sum of expected discrepancies, over all $t$ can be bounded by

$$\mathsf{DR}(\mathsf{PROP}) \leq \mathcal{O}\left( (\mathsf{ROB} - 1)T \right) + L_C C_s \sum_{t \in [T]} \mathbb{E}_{P,\pi}\left[ (\lambda\gamma)^p \right]^{1/p}$$

$$\leq \mathcal{O}\left( (\mathsf{ROB} + \lambda\gamma - 1)T \right). \tag{32}$$

Combining (31) and (32), we complete the proof.

## C.2   Proof of Theorem 5.2

Let $\mathcal{MDP}$ be the set of all MDP models $\mathsf{MDP}(\mathcal{X}, \mathcal{U}, T, P, c)$ satisfying Assumption 1,2, and 3. To prove Theorem 5.2, noting that by the definitions of consistency and robustness, we apply Theorem 5.1 to derive a bound on the worst-case ratio of expectations:

$$\sup_{\mathcal{MDP}} \mathsf{RoE}(\varepsilon) \leq 1 + \sup_{\mathcal{MDP}} \frac{\mathsf{DR}(\mathsf{PROP})}{J^\star} \leq \min\left\{ 1 + \mathcal{O}\left( \frac{\varepsilon}{T} \right) + \mathcal{O}((1-\lambda)\gamma), \mathsf{ROB} + \mathcal{O}(\lambda\gamma) \right\},$$

which implies that PROP with the BLACK-BOX Procedure is $(1 + \mathcal{O}((1-\lambda)\gamma))$-consistent and $(\mathsf{ROB} + \mathcal{O}(\lambda\gamma))$-robust.

## C.3   Proof of Theorem 5.3

*Proof.* According to Lemma 3, the expected dynamic regret of PROP satisfies

$$\mathsf{DR}(\mathsf{PROP}) = \sum_{t \in [T]} \mathbb{E}_{P,\pi}\left[ Q_t^\star(x_t, u_t) - \inf_{v \in \mathcal{U}} Q_t^\star(x_t, v) \right],$$

where $\pi$ denotes PROP. For notational simplicity, we introduce the following notation:

$$\Delta Q_t^\star(P, \pi) := \mathbb{E}_{P,\pi}\left[ Q_t^\star(x_t, u_t) - \inf_{v \in \mathcal{U}} Q_t^\star(x_t, v) \right],$$

$$\Delta \widetilde{Q}_t(P, \pi) := \mathbb{E}_{P,\pi}\left[ \widetilde{Q}_t(x_t, u_t) - \inf_{v \in \mathcal{U}} \widetilde{Q}_t(x_t, v) \right].$$

With the BLACK-BOX Procedure, we set $R_t = \lambda \eta_t$ with some hyper-parameter $0 \leq \lambda \leq 1$. Therefore, there exists Lipschitz continuous Q-value predictions $(\widetilde{Q}_t : t \in [T])$ with a Lipschitz constant $L_Q$ such that

$$\mathsf{DR}(\mathsf{PROP}(\mathsf{BLACK}\text{-}\mathsf{BOX})) \geq \sum_{t \in [T]} \left( \Delta Q_t^\star(P, \pi) - \Delta \widetilde{Q}_t(P, \pi) + (1-\lambda)L_Q\gamma \right). \tag{33}$$

First, we verify that Wasserstein robust policies exist since we can construct a transition probability $P$ such that the states in different times are independent. Denote by OPT the expected optimal total cost. We can construct cost functions $(c_t : t \in [T])$ that are Lipschitz continuous with a Lipschitz constant $L_c$ and Q-advice $(\widetilde{Q}_t : t \in [T])$ satisfying

$$\frac{\sum_{t \in [T]} \left( \Delta Q_t^\star(P, \pi) - \Delta \widetilde{Q}_t(P, \pi) \right)}{\mathsf{OPT}} \geq \Omega \left( \mathsf{ROB} + \frac{\lambda \gamma L_c}{\mathsf{OPT}} T \right).$$

Note that the corresponding Q-value predictions satisfy Assumption 3. Let $\mathcal{MDP}$ be the set of all MDP models $\mathsf{MDP}(\mathcal{X}, \mathcal{U}, T, P, c)$ satisfying Assumption 1,2, and 3. Combining above with (33), and noting that in Assumption 1, $c_t(x, u) > 0$ for all $t \in [T]$, $x \in \mathcal{X}$, and $u \in \mathcal{U}$, for any $\varepsilon \geq 0$, the ratio of expectations can be bounded by

$$\mathsf{RoE}(\mathsf{PROP}) = 1 + \sup_{\mathcal{MDP}} \frac{\mathsf{DR}(\mathsf{PROP})}{\mathsf{OPT}} = 1 + \Omega \Big( (1 - \lambda) L_Q \gamma + \min\{\varepsilon, \lambda \gamma L_c + \mathsf{ROB}\} \Big),$$

which implies that PROP cannot be both $(1 + o(\lambda \gamma))$-consistent and $(\mathsf{ROB} + o((1 - \lambda)\gamma))$-robust for any $0 \leq \lambda \leq 1$. $\qquad \square$

## D    Grey-Box Consistency and Robustness Analysis

In the following, we present a dynamic regret bound for the grey-box setting (Section 4.2) that is analogous to the one presented in Theorem 5.1 for the black-box scenario.

First, in addition to Definition 4, we further recall the following quantities used in Lemma 4 for notational convenience:

$$\zeta_t^Q (x_t, u_t) := \widetilde{Q}_t (x_t, u_t) - Q_t^\star (x_t, u_t), \tag{34}$$

$$\zeta_t^V (x_t) := \inf_{v \in \mathcal{U}} \widetilde{Q}_t (x_t, v) - \inf_{v \in \mathcal{U}} Q_t^\star (x_t, v), \tag{35}$$

where by definition, $\zeta_t^Q$ and $\zeta_t^V$ depend on the random trajectory $((x_t, u_t) : t \in [T])$. Denote $\mu_t := \zeta_t^V - \zeta_t^Q$. Note that when the environment is stationary, under some model assumptions and with a Reproducing kernel Hilbert space (RKHS) being the function class, the optimism lemma (Lemma 5.2) in [6] shows that with probability at least $1 - (2T^2H^2)^{-1}$, the generated Q-value functions satisfy $\sum_{(h,t) \in [H] \times [T]} \mathbb{E}_{P, \pi} [\delta_{h,t} + \mu_{h,t}] = \widetilde{O}(H\Gamma_K(T, \lambda)\sqrt{T})$ where $H$ is the number of episodes and $\widetilde{O}(\cdot)$ omits logarithmic terms and $T\Gamma_K(T, \lambda)$ is the maximal information gain [71] that characterizes the intrinsic complexity of the function class.

Denote the by $\varphi_t$ the per-step cost difference between the robust baseline and the optimal policy at time $t \in [T]$ such that $\sum_{t \in [T]} \varphi_t = \Theta((\mathsf{ROB} - 1)T)$. Suppose the robust baseline $\overline{\pi}$ is $\gamma$-locally $p$-Wasserstein-robust. The following theorem presents a preliminary result that will be used to prove Theorem 5.4.

**Theorem D.1** (Grey-Box: Dynamic Regret). *Consider any MDP model satisfying Assumption 1,2, and 3. The expected dynamic regret of* PROP *(Algorithm 1) with the* GREY-BOX *Procedure satisfies the following bound:*

$$\mathsf{DR}(\mathsf{PROP}) \leq \sum_{t \in [T]} \min \left\{ \underbrace{\mathbb{E}_{P, \pi} [\mu_t] + L_Q \mathbb{E}_{P, \pi} (\eta_t (x_t) - R_t)}_{\text{Consistency Bound (Lemma 3)}}, \underbrace{\varphi_t + L_C C_s \mathbb{E}_{P, \pi} [(R_t)^p]^{1/p}}_{\text{Robustness Bound (Lemma 2)}} \right\}, \tag{36}$$

*where $L_Q$ and $L_C$ are Lipschitz constants, $\mu_t := \zeta_t^V - \zeta_t^Q$, and $\eta_t (x) := \|\widetilde{\pi}_t (x) - \overline{\pi}_t (x)\|_{\mathcal{U}}$.*

### D.1    Proof of Theorem D.1

A central step in the proof of Theorem D.1 is to combine the per-step analysis in the consistency and robustness results in Lemma 2 and 3 and apply the selection of budgets in (7) (see Section 4.2). Combining (22) and (23) and summing over all $t$, (36) follows.

## D.2 Proof of Theorem 5.4

**Consistency Analysis.** We first show that the PROP policy with the GREY-BOX procedure is 1-consistent. Let $\varepsilon(p, \rho) = 0$ for some $p \in [0, \infty]$ and let $\rho$ be the trajectory distribution generated by PROP (defined in (4)). Then we must have

$$\mathbb{E}_{P,\pi}\left[\zeta_t^Q\right] = \left[\widetilde{Q}_t\left(x_t, u_t\right) - Q_t^\star\left(x_t, u_t\right)\right] = 0,$$

$$\mathbb{E}_{P,\pi}\left[\zeta_t^V\right] = \mathbb{E}_{P,\pi}\left[\inf_{v \in \mathcal{U}} \widetilde{Q}_t\left(x_t, v\right) - \inf_{v \in \mathcal{U}} Q_t^\star\left(x_t, v\right)\right] = 0,$$

for any $t \in [T]$. Consider the expectation of the TD-error (with randomness taken over the action and state trajectories). From Lemma 4 we know $\mathbb{E}_{P,\pi}[\delta_t\left(u_{t-1}, x_{t-1}, x_t\right)] = \mathbb{E}_{P,\pi}[\zeta_t^V(x_t) - \zeta_{t-1}^Q(x_{t-1}, u_{t-1})]$. This implies that the TD-error $\delta_t$ must satisfy

$$\mathbb{E}_{P,\pi}\left[\delta_t\right] = \mathbb{E}_{P,\pi}[\zeta_t^V - \zeta_{t-1}^Q] = 0. \tag{37}$$

Similarly, we have

$$\mathbb{E}_{P,\pi}\left[\mu_t\right] = \mathbb{E}_{P,\pi}\left[\zeta_t^V - \zeta_t^Q\right] = 0.$$

Therefore, by the construction of the robustness budget $R_t$ in (7) for the GREY-BOX Procedure,

$$\mathbb{E}_{P,\pi}\left[\eta_t - R_t\right] \le \mathbb{E}_{P,\pi}\left[\frac{\beta}{L_Q} \sum_{s=1}^t \delta_s\right] = 0.$$

Applying Theorem D.1, we get when $\varepsilon = 0$ (i.e., the machine-learned policy $\widetilde{\pi}$ is optimal), then $\mathsf{RoE}(0) = 1$, implying that PROP is 1-consistent.

**Robustness Analysis.** By Lemma 4, we get for any trajectory $((x_t, u_t) : t \in [T])$ and $t \in [T]$, $\mu_t - \delta_t = \zeta_{t-1}^Q - \zeta_t^Q$. Therefore, denoting by $\zeta_{-1}^Q = 0$,

$$\sum_{s=0}^t \left(\mu_s - \delta_s\right) = \sum_{s=0}^t \left(\zeta_{s-1}^Q - \zeta_s^Q\right) = \zeta_t^Q.$$

According to Assumption 3, there exist $\Delta = o(T)$ such that $|\zeta_t^Q| \le \Delta$ for all $t \in [T]$. We consider two cases. First, consider an event $\sum_{t \in [T]} \mu_t \le \Delta$. Let $\mathcal{MDP}$ be the set of all MDP models $\mathrm{MDP}(\mathcal{X}, \mathcal{U}, T, P, c)$ satisfying Assumption 1,2, and 3. Then, applying Theorem D.1,we derive a bound on the worst-case ratio of expectations:

$$\sup_{\varepsilon \ge 0} \sup_{\mathcal{MDP}} \mathsf{RoE}(\varepsilon) \le 1 + \sup_{\varepsilon \ge 0} \sup_{\mathcal{MDP}} \frac{\mathrm{DR}(\mathrm{PROP})}{J^\star} \le 1 + \mathcal{O}\left(\frac{\Delta}{T}\right) = 1 + o(1) \le \mathsf{ROB} + o(1).$$

Now, if $\sum_{t \in [T]} \mu_t > \Delta$, then we must have $\sum_{t \in [T]} \delta_t > 0$. Since the action space is compact, $\eta_t \le \gamma$, which is bounded. There exists some hyper-parameter $\beta > 0$ such that $R_t = 0$. Therefore, the PROP with the GREY-BOX Procedure will be switched to the robust baseline $\overline{\pi}$. Without loss of generality, assume $0 \le m < T$ is the largest time index such that $\sum_{s=1}^m \mu_s > \Delta$ and $\sum_{s=1}^{m-1} \mu_s \le \Delta$. Applying Theorem D.1, we have

$$\mathrm{DR}(\mathrm{PROP}) \le \sum_{s=1}^{m-1} \mu_s + \sum_{s=m}^{T-1} \varphi_s + \mathcal{O}(\eta_m) \le \sum_{s=1}^{m-1} \mu_s + \sum_{s=m}^{T-1} \varphi_s + \mathcal{O}(\gamma),$$

implying that

$$\sup_{\varepsilon \ge 0} \sup_{\mathcal{MDP}} \mathsf{RoE}(\varepsilon) \le \mathsf{ROB} + \mathcal{O}\left(\frac{\Delta + \gamma}{T}\right) \le \mathsf{ROB} + o(1).$$

# E  Proof of Theorem A.1

To show Theorem A.1, we first show a technical lemma with respect to $\mathsf{MPC}_T$, which plans until the end of the episode from the first time step.

**Lemma 5.** *Suppose Assumptions 4 and 5 hold. For each step $t \in [T]$, the control policy of $\mathsf{MPC}_T$ can be rewritten as $u_t = \overline{K}_t x_t$, for some matrices $(\overline{K}_t : t \in [T])$ satisfy that $\|\overline{K}_t\| \leq C$ for all $t \in [T]$, and*

$$\left\|(A_{t'-1} + B_{t'-1}\overline{K}_{t'-1}) \cdots (A_t + B_t\overline{K}_t)\right\| \leq C\lambda^{t'-t}, \forall t, t' \in [T], \ t' \geq t,$$

*where $\lambda, C$ are as defined in Theorem A.1.*

*Proof.* To simplify the notation, we define

$$\Gamma_{t,t'} = \begin{cases} \text{diag}\left(Q_t, R_t, \ldots, R_{t'-1}, Q_{t'}\right) & \text{if } t' = T-1 \\ \text{diag}\left(Q_t, R_t, \ldots, R_{t'-1}, P_{t'}\right) & \text{otherwise} \end{cases}. \tag{38}$$

By the KKT conditions, we see that for any $t \in [T]$, the predictive optimal solution $\psi_{t,T-1}(x_t)$ is given by

$$\begin{pmatrix} x_{t|t} \\ u_{t|t} \\ \vdots \\ \dfrac{x_{T-1|t}}{\eta_{t|t}} \\ \vdots \\ \eta_{T-1|t} \end{pmatrix} = \left( \begin{array}{c|c} \Gamma_{t,T-1} & (\Xi_{t,T-1})^\top \\ \hline \Xi_{t,T-1} & \end{array} \right)^{-1} \begin{pmatrix} 0 \\ \vdots \\ 0 \\ x_t \\ 0 \\ \vdots \\ 0 \end{pmatrix}. \tag{39}$$

Therefore, $u_{t|t}$ is a linear function of $x_t$, and this relationship defines $\overline{K}_t$. Lemma G.2 in [19] implies that $\|\overline{K}_t\| \leq C$. Note that the block matrix is invertible since $(Q_t : t \in [T])$ and $(R_t : t \in [T])$ are positive definite.

To simplify the notation, we define the state transition matrix

$$\Phi_{t,t'} := (A_{t'-1} + B_{t'-1}\overline{K}_{t'-1}) \cdots (A_t + B_t\overline{K}_t).$$

Consider an arbitrary state $x_t$. Note that $\left(x_{t'|t} : t \leq t' < T\right)$ is the optimal trajectory when there is no disturbance after step $t$. By the principle of optimality, we see that $\left(x_{t'|t} : t \leq t' < T\right)$ is identical with the actual trajectory of $\mathsf{MPC}_T$ after step $t$. In other words, for arbitrary $x_t$, the multi-step transition matrix $\Phi_{t,t'}$ satisfies

$$x_{t'|t} = \Phi_{t,t'} x_t.$$

Lemma G.2 in [19] implies that $\|\Phi_{t,t'}\| \leq C\lambda^{t'-t}$. $\qquad\square$

Lemma 5 shows that $\mathsf{MPC}_T$ has the same effect as a time-varying linear feedback controller that is exponentially stable. We generalize this property to $\mathsf{MPC}_k$ with a smaller prediction horizon (Lemma 6) by showing that $\mathsf{MPC}_k$ behaves similar to $\mathsf{MPC}_T$ when $k$ is sufficiently large.

**Lemma 6.** *Suppose Assumptions 4 and 5 hold. Let $(C, \lambda)$ be the same as Lemma 5. For each step $t \in [T]$, the control policy of $\mathsf{MPC}_k$ can be rewritten as $u_t = K_t^k x_t$, for some matrices $\{K_t^k\}_{t \in [T]}$ satisfy that*

$$\left\|K_t^k\right\| \leq C, \text{ and } \left\|K_t^k - \overline{K}_t\right\| \leq C^2 a \cdot \lambda^{2k}.$$

*Further, for any $\widehat{\lambda} > \lambda$, when $k \geq \min\{T, \frac{1}{2}\log\left(C^3 ba\lambda/(\widehat{\lambda} - \lambda)\right)/\log(1/\lambda)\}$, we have*

$$\left\|(A_{t'-1} + B_{t'-1}K_{t'-1}^k) \cdots (A_t + B_t K_t^k)\right\| \leq C\widehat{\lambda}^{t'-t}, \text{ for any } t, t' \in [T], t' \geq t.$$

*Proof.* Let $\bar{t} := \min\{t+k, T-1\}$. By the KKT conditions, we see that for any $t \in [T]$, the predictive optimal solution $\psi_{t,\bar{t}}(x_t; P_{\bar{t}})$ is given by

$$
\begin{pmatrix} x_{t|t} \\ u_{t|t} \\ \vdots \\ x_{\bar{t}|t} \\ \hline \eta_{t|t} \\ \vdots \\ \eta_{\bar{t}|t} \end{pmatrix} = \left( \begin{array}{c|c} \Gamma_{t,\bar{t}} & (\Xi_{t,\bar{t}})^\top \\ \hline \Xi_{t,\bar{t}} & \end{array} \right)^{-1} \begin{pmatrix} 0 \\ \vdots \\ 0 \\ x_t \\ 0 \\ \vdots \\ 0 \end{pmatrix}. \tag{40}
$$

Therefore, $u_{t|t}$ is a linear function of $x_t$, and this relationship defines $K_t^k$. By Lemma G.2 of [19], we see that $\left\| K_t^k \right\| \le C$.

When $\bar{t} < T-1$, construct an auxiliary disturbance sequence $\widehat{w}_{t:T-2|t}$ with $\widehat{w}_{\bar{t}|t} := -A_t \psi_{t,\bar{t}}(x_t)$ and $\widehat{w}_{t'|t} = 0$ for all $t' \ne \bar{t}$. We see that

$$
\psi_{t,\bar{t}}(x_t)[u_{t|t}] = \psi_{t,T-1}(x_t, \widehat{w}_{t:T-2|t}; Q_{T-1})[u_{t|t}].
$$

Therefore, we see that

$$
\begin{aligned}
& \left\| \psi_{t,\bar{t}}(x_t)[u_{t|t}] - \psi_{t,T-1}(x_t)[u_{t|t}] \right\| \\
&= \left\| \psi_{t,T-1}(x_t, \widehat{w}_{t:T-2|t}; Q_{T-1})[u_{t|t}] - \psi_{t,T-1}(x_t, 0_{\times(T-t-1)}; Q_{T-1})[u_{t|t}] \right\| \\
&\le C\lambda^k \left\| \widehat{w}_{\bar{t}|t} \right\| \tag{41a} \\
&\le C^2 a \cdot \lambda^{2k} \left\| x_t \right\|, \tag{41b}
\end{aligned}
$$

where we have applied the perturbation bounds in Lemma G.2 of [19] in (41a) and (41b). Since this inequality holds for any arbitrary $x_t$, we see that $\left\| K_t^k - \overline{K}_t \right\| \le C^2 a \cdot \lambda^{2k}$. To simplify the notation, we denote $\epsilon := C^2 a \cdot \lambda^{2k}$.

We can derive the following bound in terms of the $\ell_2$ norm:

$$
\begin{aligned}
& \left\| (A_{t'-1} + B_{t'-1} K_{t'-1}^k) \cdots (A_t + B_t K_t^k) \right\| \\
&\le \sum_{j=0}^{t'-t} \binom{t'-t}{j} C^{j+1} \lambda^{t-t'} (b\epsilon)^j \tag{42a} \\
&= C\lambda^{t'-t} (1 + Cb\epsilon)^{t'-t} \\
&\le C\widehat{\lambda}^{t'-t}, \tag{42b}
\end{aligned}
$$

where we use the decomposition that for any $t'' \in \{t, \ldots, t'-1\}$,

$$
A_{t''} + B_{t''} K_{t''}^k \le (A_{t''} + B_{t''} \overline{K}_{t''}) + B_{t''}(K_{t''}^k - \overline{K}_{t''})
$$

and $\left\| B_{t''}(K_{t''}^k - \overline{K}_{t''}) \right\| \le b\epsilon$ in (42a). We also use Lemma 5 in (42a) and the assumption that

$$
k \ge \frac{1}{2} \log \left( C^3 ba\lambda/(\widehat{\lambda} - \lambda) \right) / \log(1/\lambda)
$$

in (42b). $\qquad\square$

To establish a dynamic regret bound that depends on the offline optimal cost, we first need to show a lower bound of $J^*$ that depends on the "power" of the unknown disturbances.

**Lemma 7.** *The offline optimal cost is lower bounded by*

$$
J^* \ge \frac{\mu}{4(1 + a^2 + b^2)} \sum_{t=0}^{T-2} \left\| w_t \right\|^2.
$$

*Proof.* Note that the dynamics of the LTV system can be rewritten as

$$x_{t+1} - A_t x_t - B_t u_t = w_t.$$

Taking norms on both sides of the equality gives

$$\|w_t\| = \|x_{t+1} - A_t x_t - B_t u_t\|$$

$$\leq \|x_{t+1}\| + \|A_t x_t\| + \|B_t u_t\| \tag{43a}$$

$$\leq \|x_{t+1}\| + a \|x_t\| + b \|u_t\|, \tag{43b}$$

where we have used the triangle inequality in (43a) and the definition of the induced matrix $\ell_2$-norm in (43b). Taking the squares of both sides and applying the Cauchy-Schwartz inequality together imply

$$\|w_t\|^2 \leq (\|x_{t+1}\| + a \|x_t\| + b \|u_t\|)^2 \leq \frac{1 + a^2 + b^2}{\mu} \left( \mu \|x_{t+1}\|^2 + \mu \|x_t\|^2 + \mu \|u_t\|^2 \right). \tag{44}$$

By (44) and the assumptions on $Q_t$ and $R_t$, we obtain that

$$\frac{\mu}{2(1 + a^2 + b^2)} \cdot \sum_{t=0}^{T-1} \|w_t\|^2 \leq \frac{1}{2} \sum_{t=0}^{T-2} \left( \mu \|x_{t+1}\|^2 + \mu \|x_t\|^2 + \mu \|u_t\|^2 \right)$$

$$\leq 2 \sum_{t=0}^{T-1} c_t(x_t, u_t).$$

Since the above inequality holds for any arbitrary trajectory $((x_t, u_t) : t \in [T])$, we conclude that Lemma 7 holds. $\qquad\square$

Since the Wasserstein robustness (see Definition 3) is for distributions on the state-action space, we also prove a technical lemma below that helps convert the contraction on deterministic state/action pairs to the contraction on distributions. Let $W_1(\mu, \nu)$ denote the Wasserstein 1-distance between two distributions $\mu$ and $\nu$.

**Lemma 8.** *Suppose $\varphi : \mathcal{Y} \to \mathcal{W}$ is a deterministic function that satisfies $\|\varphi(v) - \varphi(v')\|_{\mathcal{W}} \leq \kappa \|v - v'\|_{\mathcal{Y}}$ for any $v, v' \in \mathcal{Y}$. Then, for any pair of distributions $\rho$ and $\rho'$ on $\mathcal{Y}$, we have $W_1(\varphi(\rho), \varphi(\rho')) \leq \kappa W_1(\rho, \rho')$.*

*Proof.* Recall that $W_1(\rho, \rho') := \inf_J \int \|v - v'\|_{\mathcal{Y}} \, dJ(v, v')$, where $J$ is a joint distribution on $\mathcal{Y} \times \mathcal{Y}$ with marginals $\rho$ and $\rho'$. We define a mapping $\Phi : \mathcal{Y} \times \mathcal{Y} \to \mathcal{Z} \times \mathcal{Z}$ as $\Phi(v, v') := (\varphi(v), \varphi(v'))$. We see that $\Phi J$ gives a joint distribution on $\mathcal{W} \times \mathcal{W}$ with marginals $\varphi(\rho)$ and $\varphi(\rho')$, and it satisfies

$$\int \|u - u\|_{\mathcal{W}} \, d(\Phi J)(u, u') = \int \|\varphi(v) - \varphi(v')\|_{\mathcal{Y}} \, dJ(v, v') \leq \varepsilon \int \|v - v'\|_{\mathcal{Y}} \, dJ(v, v').$$

Note that the above inequality holds for any $J$ with marginals $\rho$ and $\rho'$. Thus Lemma 8 holds. $\qquad\square$

Now we are ready to show Theorem A.1.

For a state $x$ at time step $t \in [T]$, let $x_{t:t'}(x)$ and $u_{t:t'}(x)$ denote the corresponding state and action of MPC at time step $t'$. By Lemma 6, we see that for any state-action pairs $(x, u)$ and $(x', u')$ at step $t_1$, we have

$$\|(x_{t_1+1:t_2}, u_{t_1+1:t_2})(A_{t_1} x + B_{t_1} u + w_{t_1}) - (x_{t_1+1:t_2}, u_{t_1+1:t_2})(A_{t_1} x' + B_{t_1} u' + w_{t_1})\|$$

$$\leq (1 + C) \|x_{t_1+1:t_2}(A_{t_1} x + B_{t_1} u + w_{t_1}) - x_{t_1+1:t_2}(A_{t_1} x' + B_{t_1} u' + w_{t_1})\| \tag{45a}$$

$$\leq (1 + C) C \widehat{\lambda}^{t_2 - t_1 - 1} \|A_{t_1}(x - x') + B_{t_1}(u - u')\| \tag{45b}$$

$$\leq (1 + C) C (a + b) \widehat{\lambda}^{t_2 - t_1 - 1} \|(x, u) - (x', u')\|, \tag{45c}$$

where we have used Lemma 6 in (45a) and (45b); Moreover, we have applied the assumption that $\|A_{t_1}\| \leq a$ and $\|B_{t_1}\| \leq b$ and the triangle inequality in (45c). Since (45) establishes a contraction for a deterministic state-action pair and the dynamics is deterministic, applying Lemma 8 finishes the proof of the first conclusion of Theorem A.1.

Using a similar decomposition technique with [19], by Lemma 6, we see that the trajectory $(x_t : t \in [T])$ of PROP satisfies that

$$\|x_t\| \leq \sum_{t'=0}^{t-1} \left\| \Phi_{t',t}^k \right\| \cdot (\|w_{t'}\| + b\overline{R}) \leq C \sum_{t'=0}^{t-1} \widehat{\lambda}^{t-t'} (d + b\overline{R}) \leq \frac{C(d+b\overline{R})}{1-\widehat{\lambda}}, \tag{46}$$

where we denote $\Phi_{t',t}^k := (A_{t'-1} + B_{t'-1} K_{t'-1}^k) \cdots (A_t + B_t K_t^k)$ and the assumption that PROP deviates at most $\overline{R}$ from $\mathsf{MPC}_k$'s action. We also see that

$$\|u_t\| \leq \left\| K_t^k x_t \right\| + \overline{R} \leq C\overline{R}_x + \overline{R}. \tag{47}$$

This finishes the proof of the second statement in Theorem A.1.

Let the trajectory of $\mathsf{MPC}_k$ when executed without the machine-learned advice be denoted by $(\overline{x}_t : t \in [T])$. We see that

$$\|\overline{x}_t\| = \left\| \sum_{t'=0}^{t-1} \Phi_{t',t}^k w_{t'} \right\| \leq C \sum_{t'=0}^{t-1} \widehat{\lambda}^{t-t'} \|w_{t'}\|.$$

Applying the Cauchy-Schwarz inequality, we obtain

$$\|\overline{x}_t\|^2 \leq \left( C \sum_{t'=0}^{t-1} \widehat{\lambda}^{t-t'} \|w_{t'}\| \right)^2 \leq C^2 \left( \sum_{t'=0}^{t-1} \widehat{\lambda}^{t-t'} \right) \left( \sum_{t'=0}^{t-1} \widehat{\lambda}^{t-t'} \|w_{t'}\|^2 \right)$$

$$\leq \frac{C^2}{1-\widehat{\lambda}} \sum_{t'=0}^{t-1} \widehat{\lambda}^{t-t'} \|w_{t'}\|^2. \tag{48}$$

For the control actions of $\mathsf{MPC}_k$, we also see that

$$\|\overline{u}_t\|^2 = \left\| K_t^k \overline{x}_t \right\|^2 \leq C^2 \|\overline{x}_t\|^2 \leq \frac{C^4}{1-\widehat{\lambda}} \sum_{t'=0}^{t-1} \widehat{\lambda}^{t-t'} \|w_{t'}\|^2. \tag{49}$$

Therefore, we get the following bound on the total cost:

$$J(\mathsf{MPC}_k) = \sum_{t=0}^{T-1} \left( \frac{1}{2} (\overline{x}_t)^\top Q_t \overline{x}_t + \frac{1}{2} (\overline{u}_t)^\top R_t \overline{u}_t \right)$$

$$\leq \frac{\ell}{2} \sum_{t=0}^{T-1} \left( \|\overline{x}_t\|^2 + \|\overline{u}_t\|^2 \right) \tag{50a}$$

$$\leq \frac{C^2(1+C^2)}{2(1-\widehat{\lambda})} \sum_{t=0}^{T-1} \sum_{t'=0}^{t-1} \widehat{\lambda}^{t-t'} \|w_{t'}\|^2 \tag{50b}$$

$$\leq \frac{C^2(1+C^2)}{2(1-\widehat{\lambda})^2} \sum_{t=0}^{T-2} \|w_t\|^2,$$

where we have used the assumption that $Q_t \preceq \ell I$ and $R_t \preceq \ell I$ in (50a); we have also used the inequalities (48) and (49) in (50b). Combining (50) with the lower bound of $J^*$ in Lemma 7 finishes the proof of the third Statement in Theorem A.1.

## F   Proof of Theorem A.2

Before showing Theorem A.2, we first state a technical lemma that establishes the relationship between the TV distance and the Wasserstein distance.

**Lemma 9.** *For any distributions $\mu, \nu$ on $\mathcal{X}$, we have*

$$W_1(\mu, \nu) = 2\mathsf{TV}(\mu, \nu) = \|\mu - \nu\|_1.$$

*Proof.* To see this, note that since $\|x - x'\|_1 = 2$ for any $x \neq x'$, the Wasserstein 1-distance $W_1(\mu, \nu)$ equals 2 times the probability mass we need to transport to convert $\mu$ to $\nu$. For every $i \in \{1, \dots, n\}$ such that $\mu_i > \nu_i$, we need to move out exactly $(\mu_i - \nu_i)$ from the probability mass at $e_i$ to other points $(e_j : j \neq i)$. Therefore, we must have

$$W_1(\mu, \nu) = 2 \sum_{i=1}^{n} \mathbf{1}(\mu_i > \nu_i) \cdot (\mu_i - \nu_i) = 2\mathsf{TV}(\mu, \nu) = \|\mu - \nu\|_1.$$

$\square$

Note that the MDP's transition kernel acts as a deterministic function. It maps the current state-action pair from $\mathcal{X} \times \mathcal{U}$ to the distribution of the subsequent state in $\mathcal{X}$. Hence, the current state-action distribution on $\mathcal{X} \times \mathcal{U}$ maps to a distribution on $\Delta(\mathcal{X})$. To proceed with this recursion, we require the distribution of the next state, which should be on $\mathcal{X}$. This is in contrast to needing the distribution of the distribution of the next state, which would be on $\Delta(\mathcal{X})$. Therefore, to convert the distributions on $\Delta(\mathcal{X})$ back to distributions on $\mathcal{X}$, we require the following lemma.

**Lemma 10.** *Let $\mu, \mu'$ be two distributions on $\Delta(\mathcal{X})$. It follows that $\|\mathbb{E}[\mu] - \mathbb{E}[\mu']\|_1 \leq W_1(\mu, \mu')$.*

Note that $\mathbb{E}[\mu]$ and $\mathbb{E}[\mu']$ are distributions on $\mathcal{X}$.

*Proof.* By the definition of the Wasserstein distance, we have

$$W_1(\mu, \mu') = \inf_J \int \|x - y\|_1 \, \mathrm{d}J(x, y),$$

where $J$ is a joint distribution on $\Delta(\mathcal{X}) \times \Delta(\mathcal{X})$ with marginals $\mu$ and $\mu'$. For any such joint distribution $J$, we have

$$\int \|x - y\|_1 \, \mathrm{d}J(x, y) \geq \left\| \int (x - y) \mathrm{d}J(x, y) \right\|_1 = \|\mathbb{E}[\mu] - \mathbb{E}[\mu']\|_1.$$

This finishes the proof of Lemma 10. $\square$

We now resume our discussion with the proof of Theorem A.2.

Given the state-action distribution $\rho$ at step $t$, let $\mu_{t:t'}(\rho)$ denote the resulting state distribution at step $t'$. We slightly abuse the notation so that for any pair $(x, u) \in \mathcal{X} \times \mathcal{U}$, $\mu_{t:t'}((x, u))$ still outputs the resulting state distribution at step $t'$. We see that

$$\|\mu_{t:t+1}((x, u)) - \mu_{t:t+1}((x', u'))\|_1 \leq \|x - x'\|_1 + \|u - u'\|_1.$$

Therefore, by Lemmas 8 and 10, we see that

$$W(\mu_{t_1:t_1+1}(\rho), \mu_{t_1:t_1+1}(\rho')) \leq W(\rho, \rho'). \tag{51}$$

Note that Lemma 1 and Lemma 9 imply that

$$W(\mu_{t_1:t_2}(\rho), \mu_{t_1:t_2}(\rho')) \leq \lambda^{t_2 - t_1 - 1} W(\mu_{t_1:t_1+1}(\rho), \mu_{t_1:t_1+1}(\rho')).$$

Combining this with (51) gives that

$$W(\mu_{t_1:t_2}(\rho), \mu_{t_1:t_2}(\rho')) \leq \lambda^{t_2 - t_1 - 1} W(\rho, \rho'). \tag{52}$$

For any distributions $\mu, \mu'$ on $\mathcal{X}$, we also see that $\|\bar{\pi}_{t_2}(\mu) - \bar{\pi}_{t_2}(\mu')\|_1 \leq \|\mu - \mu'\|$, which implies

$$W(\bar{\pi}_{t_2}(\mu), \bar{\pi}_{t_2}(\mu')) \leq W(\mu, \mu').$$

Substituting this into (52) gives that

$$W(\rho_{t_1:t_2}(\rho), \rho_{t_1:t_2}(\rho')) \leq 2\lambda^{t_2 - t_1 - 1} W(\rho, \rho'),$$

validating that the Wasserstein robustness (Definition 3) is satisfied.