# OpenReview forum: "Beyond Black-Box Advice: Learning-Augmented Algorithms for MDPs with Q-Value Predictions"
_NeurIPS.cc/2023/Conference — NeurIPS 2023 poster_

### Official Review · Reviewer_qteD · 2023-07-01

**Soundness:** 4 excellent
**Presentation:** 4 excellent
**Contribution:** 3 good
**Rating:** 7
**Confidence:** 3

**Summary:**

This work considers the setting of using advice in Markov Decision Processes (MDPs). They consider the setting of using only action suggestions (termed black-box) or action and Q-values (grey-box) from a potentially erroneous advisor. They propose an algorithm that decides between the advice and a robust policy by projecting the advisor’s suggested action onto a ball centered around the robust policy’s action. Depending on the type of advisor, the radii of the projection ball can be static (black-box) or dynamically-chosen based on the estimated error of the Q-values (grey-box). The algorithm is analyzed along the axes of consistency (performance) and robustness, and it is shown that the grey-box setting can achieve much better consistency and robustness than the black-box setting.

**Strengths:**

This paper is very well-written and clear. There is a lot of effort in explaining the setting, the definitions, and the approach which helps with readability and understanding.

The results are more general than prior works, which had similar results in the tabular setting (Golowich, 2021), however, I am also not an expert in this area.

The Projection Pursuit Policy (PROP) approach generalizes the black-box and grey-box settings nicely, and the results for the grey-box setting are promising for future work.


**Weaknesses:**

I think the paper overall could benefit from some empirical or practical demonstration of the setting, similar to what is presented in Appendix A, but with more focus on the applicability of the algorithm.  Ideally this would be in the main body but I understand that this is difficult given the page limit.

My only concern is in the usefulness of the setting and some of its assumptions, which I explain below and hope the authors can comment on.

An example of a setting this could be used in: we have learned an optimal or close-to-optimal policy/value function but our environment is non-stationary and may have shifted since we learned this policy. We also have a baseline policy that is known to be robust in this environment. We then use PROP to take an action depending on how much to trust the robust vs the previously-learned policy. My concerns are:

- The grey-box advice is in the form of Q-values: The choice of $(\infty, \epsilon)$-consistent policies is quite strong (defined in L243). In most realistic situations, we might only be able to guarantee the Q-function on certain data-dependent distributions. I think a more natural assumption would be to consider slowly-varying dynamics, for example where the change in $\mathbb{P}_t$ is bounded under some norm. Could you comment on the limitations of this assumption?

- Access to a robust policy: This was discussed for the case of discrete MDPs and LQR, however for the case of general MDPs, how can this policy be obtained? If the environment is time-varying, it might also affect the robustness guarantees of the policy.

- This may be out-of-scope but a central assumption is that advice is provided at each step: advice might generally only be provided in some steps and not in others but the agent should still have the ability to learn the best policy in those situations.


**Questions:**

Minor/Typos:

- Algorithm 1 box line 5 and 6 should be switched, since we don’t know all $x_t$ before taking the selected actions.

- L302, instead of $\tilde{\pi}_t$ should be $\tilde{u}_t$?


**Limitations:**

They have addressed some limitations, but some questions remain (See Weaknesses)

---

> ### Author Rebuttal · Authors · 2023-08-07
>
> Thank you so much for your insightful comments! Below are our responses to your specific comments and questions:
>
> 1. **Non-stationary Environment**:
>
> Than you so much for providing more insights and potential application domains of our methods. In fact,
> existing algorithms for non-stationary RL primarily focus on adapting classic stationary RL algorithms through methods such as slicing windows [Gajane et al.] [Cheung et al.] [Fei et al.] [Chen el al.], periodic restarts [Touati et al.] [Mao et al.], kernel-based design [Domingues et al.], and black-box reduction (MALG) [Wei et al.]. However, these algorithms have dynamic regret bounds that deteriorate as the environment changes faster.
>
> Next, we address the detailed concerns on the usefulness of the setting and assumptions below.
>
> 2. **$(\infty,\varepsilon)$-Consistent Policies**:
>
> The assumption of $(\infty,\varepsilon)$-consistent policies is stronger than assuming the norm can be bounded under a specific distribution $\rho$ for general $L_{p,\rho}$ norms (as defined in (4), line 239-240). The reason we choose to present our main results, such as Theorem 5.1 assuming the machine-learned policy is $(\infty,\varepsilon)$-consistent is to simplify the statement, since the goal of defining a consistency error is to quantify how the Q-value advice is close to optimal Q-value functions as in the traditional consistency and robustness metrics (c.f. references [2,3,19,20] in the manuscript).
>
> In fact, the only place where we need a stronger bound characterizing the Q-value error is the consistency analysis in the proof of Theorem C.1. (see line 734 in Appendix C.1), where we need to bound $\mu_t=\zeta_t^{V}-\zeta_t^{Q}$, which is defined as the difference $\widetilde{Q}_t (x_t,\widetilde{u}_t)-Q_t^{\star}(x_t,u_t^{\star}) - (\widetilde{Q}_t (x_t,u_t)-Q_t^{\star} (x_t,u_t))$.
> If we assume that the policy is $(p,\rho,\varepsilon)$-consistent  where $\rho$ is the trajectory distribution generated by the policy in Algorithm 1 (PROP), then the result also holds.
>
> Besides, it is worth mentioning that Theorem 5.4 can be proved for the general consistency error defined in (4) as well, as in the consistency analysis in Appendix D.2, if $\varepsilon(p,\rho)=0$ for the distribution $\rho$ that is the trajectory distribution generated by the policy in  Algorithm 1 (PROP)), it implies that both $\mathbb{E}[\zeta_t^{Q}]$ and $\mathbb{E}[\zeta_t^{V}]$ are zeros for any $t\in [T]$. We will further clarify this in the revised paper.
>
> 3. **Access to a Robust Policy and General MDPs**:
>
> We appreciate the reviewer's concern about obtaining a robust policy for general MDPs. Our main focus is on the general setting, and we believe that by formulating and proving our theorems for this broader context, we can significantly extend the application domains of our proposed methods. While we discussed specific cases of discrete MDPs and LQR, our theoretical framework is applicable to a wide range of MDPs, making it more versatile and relevant to various real-world scenarios.
>
> Regarding the concern about time-varying environments, we would like to clarify that our model is designed as a single-trajectory MDP. In this formulation, the transition probability $P_t$ can change over time, allowing for non-stationary MDPs. Despite that the MDP may be non-stationary, our results still hold for each episode (single-trajectory). This characteristic of our model and results ensures that the proposed approach remains applicable even in dynamic environments where the underlying dynamics may change over time.
>
> Moreover, we have provided examples of specific models with rich applications where robust policies do exist and can be constructed explicitly. These examples demonstrate the practical feasibility and relevance of our approach in settings where robust policies can be readily obtained.
>
> 4. **Assumption of Advice at Each Step**:
>
> Thank you for raising this interesting point! While our current work focuses on the assumption of advice at each step, we acknowledge the importance of exploring the new setting where advice is intermittently provided. We believe this is a valuable direction for future research, and we are more than willing to investigate and extend our approach to accommodate such scenarios.
>
> 5. **Minor/Typos**:
>
> Thank you so much for pointing out the typos in the Algorithm 1, line 5-6 and line 302! We have fixed the typos in the manuscript.
>
> **References**
>
> [Gajane et al.] Gajane, Pratik, Ronald Ortner, and Peter Auer. "A sliding-window algorithm for markov decision processes with arbitrarily changing rewards and transitions. 2018
>
> [Cheung et al.] Cheung, Wang Chi, David Simchi-Levi, and Ruihao Zhu. "Reinforcement learning for non-stationary markov decision processes: The blessing of (more) optimism." International Conference on Machine Learning. PMLR, 2020.
>
> [Fei et al.] Fei, Yingjie, et al. "Dynamic regret of policy optimization in non-stationary environments." Advances in Neural Information Processing Systems 33 (2020): 6743-6754.
>
> [Chen et al.] Chen, Liyu, and Haipeng Luo. "Near-optimal goal-oriented reinforcement learning in non-stationary environments." Advances in Neural Information Processing Systems 35 (2022): 33973-33984.
>
> [Touati et al.] Touati, Ahmed, and Pascal Vincent. "Efficient learning in non-stationary linear markov decision processes." 2020
>
> [Mao et al.] Mao, Weichao, et al. "Near-optimal model-free reinforcement learning in non-stationary episodic mdps." International Conference on Machine Learning. PMLR, 2021.
>
> [Domingues et al.] Domingues, Omar Darwiche, et al. "A kernel-based approach to non-stationary reinforcement learning in metric spaces." International Conference on Artificial Intelligence and Statistics. PMLR, 2021.
>
> [Wei et al.] Wei, Chen-Yu, and Haipeng Luo. "Non-stationary reinforcement learning without prior knowledge: An optimal black-box approach." Conference on learning theory. PMLR, 2021.

---

> > ### Comment · Reviewer_qteD · 2023-08-18
> >
> > Thank you for the thorough response, that clears up my questions and I will raise my score. After reading the discussions with other reviewers, I think that as part of future work, the practicality of the approach should be demonstrated in empirical settings. Nevertheless, in its current form, the work is a good contribution both algorithmically and theoretically.

---

> > > ### Author Response · Authors · 2023-08-18
> > >
> > > Thank you again for your constructive engagement! We're delighted that our responses have been helpful in clarifying the questions. We are actively working on demonstrating the practicality of our approach through empirical validations. If you have any further thoughts, questions, or suggestions in the future, please don't hesitate to share them.

---

### Official Review · Reviewer_BJJd · 2023-07-05

**Soundness:** 4 excellent
**Presentation:** 3 good
**Contribution:** 4 excellent
**Rating:** 7
**Confidence:** 3

**Summary:**

The authors study the tradeoff between consistency and robustness in the context of a single-trajectory time-varying Markov Decision Process (MDP) with untrusted machine-learned advice. Time-varying MDPs are relevant when dealing with environments where costs, rewards, and transition probabilities might change over time. In such settings, it is essential to design algorithms that can adapt to the changing dynamics of the environment. The proposed Projection Pursuit Policy (PROP) aims to balance the performance of an untrusted machine-learned policy with a trusted robust policy, taking into account the time-varying nature of the problem. The algorithm is designed to make better use of Q-value advice from the untrusted policy while maintaining worst-case performance guarantees close to those provided by the robust policy. The authors provide theoretical analysis for the consistency and robustness of the proposed algorithm. In the appendix, the authors also provide examples to demonstrate the versatility and effectiveness of their findings to various real-world problems.

**Strengths:**

Originality:

The paper presents a novel learning-augmented algorithm (PROP) for MDPs with Q-value predictions.

The work considers both black-box and grey-box settings, demonstrating the benefits of leveraging additional structural information in achieving improved performance guarantees.

Quality:

The submission is technically sound, and the claims are well supported by theoretical analysis.

The authors provide a thorough analysis of the consistency and robustness tradeoffs in the context of MDPs.


Clarity:

The submission is clearly written and well-organized.

Significance:

The work advances the state of the art in learning-augmented algorithms for MDPs and demonstrates the potential benefits of utilizing value-based policies as advice.

The findings are relevant to researchers and practitioners interested in developing algorithms that can adapt to changing environments while maintaining performance guarantees.

**Weaknesses:**

Insufficient comparison to existing approaches: The paper does not provide a detailed comparison of the proposed PROP algorithm with existing learning-augmented algorithms, particularly in the context of discrete action spaces. A thorough comparison, both in terms of theoretical analysis and empirical evaluations, would help demonstrate the advantages and potential improvements offered by the proposed approach over existing methods. This would also provide a clearer understanding of the practical implications and the extent to which the proposed method advances the state of the art.

**Questions:**

The authors mentioned in the abstract that their method can be applied in the discrete action space setting. However, the Projection Pursuit Policy may require adjustments to ensure that the resulting action remains feasible. When combining actions from different policies or projecting them onto a certain space, the resulting action may not lie exactly in the discrete action space. These adjustments may affect the theoretical guarantees of the algorithm, and further analysis might be needed to determine the impact of such modifications on the consistency and robustness tradeoffs of the Projection Pursuit Policy in the discrete action space setting.

The $\epsilon$ parameter in Figure 1 is not introduced. It seems that it should be the same as in Equation 4.


**Limitations:**

The dependence of the theoretical results on the parameter $\lambda$ in the black box setting may affect the practical implementation of the algorithm, as the choice of $\lambda$ could influence the trade-off between consistency and robustness. Potential strategies for selecting an appropriate value are needed.

In the grey box setting, the theoretical results are only valid for some $\beta$, which should satisfy the condition $R_t = 0$. It may be challenging to determine a suitable value of $\beta$ in practice. Further analysis of the algorithm's performance and robustness under different values of $\beta$ would be valuable.

---

> ### Author Rebuttal · Authors · 2023-08-09
>
>
> We sincerely appreciate your thoughtful feedback and valuable insights regarding our paper. Below we respond to your questions and concerns:
>
> 1. **Comparison to Existing Approaches**:
>
> >*Insufficient comparison to existing approaches: The paper does not provide a detailed comparison of the proposed PROP algorithm with existing learning-augmented algorithms, particularly in the context of discrete action spaces.*
>
> We acknowledge the importance of providing a more detailed comparison with existing learning-augmented algorithms. Our work stands as a novel contribution in the field as it represents the first-of-its-kind result in designing learning-augmented algorithms for general MDP models.
>
> Existing learning-augmented algorithms, as reviewed in the first part of Section 1.2, have mainly focused on specific models and online problems, such as ski rental, caching, bipartite matching, convex body chasing, online covering, etc. These approaches primarily revolve around black-box advice, making them more limited in their applicability to general MDP scenarios.
>
> In the revised version of our paper, we will provide a more comprehensive analysis by comparing our proposed approach against relevant existing methods to clarify the distinctive advantages and contributions of our work in the context of learning-augmented algorithms.
>
> 2. **Discrete Action Space Setting**:
>
> >*The authors mentioned in the abstract that their method can be applied in the discrete action space setting. However, the Projection Pursuit Policy may require adjustments to ensure that the resulting action remains feasible.*
>
> Thank you for the insightful question and we truly appreciate your attention to the applicability of our method in the discrete action space setting.
>
> The footnote on page 3, under line 138 briefly sketches the generalization of our model and methods to discrete settings. We acknowledge that our presentation may have appeared simplistic, and we apologize for any confusion caused. Let us elaborate how our the algorithm can be generalized. In the discrete setting, we can lift the norm definitions to distributions over $\mathcal{X}$ and $\mathcal{U}$, allowing us to define Wasserstein robustness effectively. Our results can be extended to this discrete setting as well. Specifically, when the action space $\mathcal{U}$ is discrete, we can define it as the set of all probability distributions on a finite action space, which forms a probability simplex. This ensures that $\mathcal{U}$ remains convex and compact, aligning with the requirements for our approach in the continuous action space setting to be applied.
>
> Consider an MDP environment with a finite state space $\mathcal{S}$ and a finite action space $\mathcal{A}$. Denote by $\Delta(\mathcal{A})$ and $\Delta(\mathcal{S})$ the sets of probability distributions over $\mathcal{A}$ and $\mathcal{S}$. Given a policy $\overline{\pi}_t: \mathcal{S} \to \Delta(\mathcal{A})$ for $t \in [T]$, let $\{\overline{P}_t\}$ denote the lifted state transition probability that maps $\mathcal{S}$ to $\Delta(\mathcal{S})$, which is defined as
> $\overline{P}_t(s; s') = \sum_a\overline{\pi}_t(s; a){P}_t(s, a; s').$
> We can find a Wasserstein robust baseline policy in this discrete setting. The key idea is to show that the one-step transition probability $\overline{P}_t$ forms a contractive mapping with respect to the total variance distance (Lemma 1 in Appendix A.2.) and this contraction property can be used to establish the desired Wasserstein robustness.
>  *A detailed discussion can be found in the submitted supplementary material (Appendix A.2).*  To address your concern, we will provide a more concrete and detailed explanation in the main body of the paper regarding the application of our approach in the discrete action space setting. Thank you again for your feedback!
>
> 3. **Parameter in Figure 1**:
>
> >*The parameter $\varepsilon$ in Figure 1 is not introduced. It seems that it should be the same as in Equation 4.*
>
> We apologize for causing the confusion. It should be the same as in Equation 4. We will ensure consistency in notation throughout the paper and make the necessary corrections for clarity.
>
> 4. **Practical Implementation and Parameter Selection**:
>
> >*The dependence of the theoretical results on the parameter $\lambda$ in the black box setting may affect the practical implementation. Potential strategies are needed.*
>
> Thank you for pointing out this limitation. In fact, we haven't focused too much on optimizing the choice of $\lambda$ for the black-box procedure, since the black-box procedure serves more like a benchmark (as used in previous learning-augmented algorithms) to highlight the usefulness of the "gray-box" feedback (Q-values), which is proven to nontrivially outperform the black-box setting. Our main focus is the selection and analysis of the robustness budget $R_t$ (Equation 7) in the grey-box setting.
>
> Besides, our analysis on the black-box procedure can be applied to any choice of $\lambda\in [0,1]$, as stated in Theorem C.1. in the appendix. In our revised version, we will add more clarification based on your suggestion.
>
>
> 5. **Characterize the Right Choice of $\beta$ in the Grey-Box Setting**:
>
> >*In the grey box setting, the theoretical results are only valid for some $\beta$. It may be challenging to determine a suitable value of in practice.*
>
> We appreciate your point about the challenge of determining a suitable hyperparameter $\beta$ of in the grey box setting.
>
> We agree that determining $\beta$ is of special theoretical interests, based on our current results, which validate the existence of such hyperparameters to guarantee the stated consistency and robustness tradeoff.
>
> In our future work, we will conduct further analysis to explore the algorithm's performance and robustness under different values of $\beta$ to offer valuable insights into the algorithm's behavior in practical scenarios.
>
> Thank you again for your insightful feedback!

---

> > ### Comment · Reviewer_BJJd · 2023-08-16
> >
> > Thank you for providing a thorough and well-considered rebuttal to my concerns. I appreciate the time and effort you have put into addressing my questions, and I believe they have been satisfactorily resolved.

---

### Official Review · Reviewer_SGod · 2023-07-06

**Soundness:** 3 good
**Presentation:** 2 fair
**Contribution:** 2 fair
**Rating:** 6
**Confidence:** 2

**Summary:**

The paper proposes a novel algorithm that combines a robust policy and a potentially untrusted machine learning based policy. The idea is to be able to achieve a trade-off between robustness and consistency or high performance entailed by the ML policy. The paper suggests a novel projection algorithm along with theoretical guarantees on the proposed algorithm.

**Strengths:**

The paper seems to have solid theoretical grounding on the proposed algorithm, the background problem is also relatively well motivated and can be useful for general contexts where a ML policy is present alongside a default robust policy. This can be of interest to the general decision making practitioners for real applications.

**Weaknesses:**

The paper can be potentially improved by being more concrete about the problems and applications it aims to address, as well as showcasing some empirical evidence that the suggested algorithm achieves nice guarantees predicted by the theory.

**Questions:**

#### === *Motivation* ===

Checking my understanding of the problem at hand, we have a default robust policy $\bar{\pi}$ as well as a ML based policy $\tilde{\pi}$ whose performance guarantee is potentially not trusted. The aim is to derive a new policy that combines the two and be able to achieve consistency vs. robustness trade-off in real-time decision making, does this sound right?

I'd suggest giving more concrete examples for this setup, e.g. education, medical and so on, such that it is better to contextualize the problem at hand and better understand various algorithmic components.

#### === *Fig 1* ===

Rightmost plot of Fig 1 -- what's on the x-axis and y-axis? There should be more explanations in the caption.

#### === *Projection pursuit* ===

The projection pursuit policy derived in Sec 4 seems to be the core contribution of the paper and proposes to combine the two policies using a projection step. A main criticism might be that the algorithm seems specialized to continuous action, where the projection is more well defined? What if the action space is discrete, do we carry out projection in the logits / probability space or in the discrete action space?

#### === *Empirical assessment* ===

There is no empirical assessment of the approach at all in the paper, as the narrative is purely theoretical. Though the theoretical contribution of the paper seems solid, readers from the NeurIPS community would certainly benefit from certain empirical validations of the theoretical insights in the paper. One should at least test the e.g. projection pursuit algorithm in some even potentially artificial domains, or even real domains (simulated envs) and showcase how the bound prediction is reflected in practice. One would also benefit from a discussion on how those hyper-parameters (e.g. regret budget $R_t$) would actually impact algorithm performance in practice.

**Limitations:**

Discussed above.

---

> ### Author Rebuttal · Authors · 2023-08-07
>
> Thank you for your valuable feedback and insightful comments. Below are our responses to your specific comments and questions:
>
> 1. **Motivation**:
>
> >*The aim is to derive a new policy that combines the two and be able to achieve consistency vs. robustness trade-off in real-time decision making, does this sound right? I'd suggest giving more concrete examples for this setup.*
>
> Yes, your description is correct. Indeed, the core objective of our work is to derive a new policy that effectively balances consistency and robustness in real-time decision-making scenarios.
>
> In detail, our approach involves combining a default robust policy (denoted as $\overline{\pi}$) with a machine-learning-based policy (denoted as $\widetilde{\pi}$) whose performance guarantee may not be entirely trusted.  In addition, we would like to highlight that our main contribution is not only proposing such a model, but also characterizing how Q-value functions can help improve the consistency and robustness tradeoff, a.k.a the "gray-box setting" in the manuscript. We show that this gray-box feedback has nontrivial advantages in terms of the trade-off over simply combing a black-box policy with a robust baseline.
>
> We agree with your suggestion of furnishing our models with more concrete examples to enhance the presentation. In the revised version, we will include real-world applications to better illustrate the practical relevance of our algorithm. For instance, a relevant example could be decision-making in power systems, where traditional methods as robust baselines *often exist*. In such scenarios, utilities companies may *hesitate to deploy* reinforcement learning algorithms for critical decision-making tasks, such as optimizing battery charging/discharging schedules or controlling inverters for voltage/frequency regulation.  Our provably efficient and robust policy provides a reliable solution for these applications.
>
> Another quick example is to navigate a drone in a windy day. The drone relies on a pre-trained value-based policy and a baseline controller such as PID or MPC. By characterizing how Q-value advice can improve the consistency and robustness tradeoff, our results offer provable solutions to strike the right balance between the value-based policy and the baselines. In our revised paper, we will use these examples to motivate our problem and show how our methods can be applied in a wide range of applications where a Wasserstein robust baseline exists.
>
> 2.  **Figure 1**:
>
> We apologize for not explicitly defining $\varepsilon$ and $\mathsf{RoE}$ in Fig 1. The formal definitions of $\varepsilon$ and $\mathsf{RoE}$ are in Eq.(4) (line 239-240) and Definition 2 (line 146-148). We will improve the caption by expilcity pointing to the right places of the definitions.
>
> 3. **Projection Pursuit**:
>
> >*What if the action space is discrete, do we carry out projection in the logits / probability space or in the discrete action space?*
>
> Thank you for raising this important point!
>
> We apologize for any confusion caused by the presentation in the manuscript. We have indeed provided a footnote on page 3, under line 138, briefly introducing how our model and methods can be generalized to discrete settings and leave the details in the appendix. We understand the importance of making this discussion more explicit in the main body of the paper. In the revised version, we will address this concern by providing a clearer and more detailed explanation.
>
> Specifically, when the action space $\mathcal{U}$ is discrete, we can define it as the set of all probability distributions on a finite action space, which forms a probability simplex that is convex and compact. To carry out the projection in the discrete action and state spaces, we can lift the norms from the original spaces to distributions on $\mathcal{X}$ and $\mathcal{U}$. This general norm can be used to define the Wasserstein robustness (Definition 3) in the discrete action space. Consider an MDP environment and denote by $\Delta(\mathcal{A})$ and $\Delta(\mathcal{S})$ the sets of probability distributions over $\mathcal{A}$ and $\mathcal{S}$. Given a policy $\overline{\pi}_t: \mathcal{S} \to \Delta(\mathcal{A})$ for $t \in [T]$, let $\{\overline{P}_t\}$ denote the lifted state transition probability that maps $\mathcal{S}$ to $\Delta(\mathcal{S})$, defined as  $\overline{P}_t(s; s') = \sum_a\overline{\pi}_t(s; a){P}_t(s, a; s').$ We can find a Wasserstein robust baseline in the discrete setting. The key idea is to show that $\overline{P}_t$ forms a contractive mapping with respect to the total variance distance (Lemma 1 in Appendix A.2.) and this contraction property is used to give the desired Wasserstein robustness. *A detailed discussion is in the supplementary material (Appendix A.2).* We will better clarify this in the main body of the revised paper.
>
> 4. **Empirical Assessment**:
>
> > *There is no empirical assessment of the approach at all in the paper, as the narrative is purely theoretical. Though the theoretical contribution of the paper seems solid, readers from the NeurIPS community would certainly benefit from certain empirical validations of the theoretical insights in the paper.*
>
> We appreciate your recognition of the solid theoretical contribution characterizing how Q-value functions can improve the consistency and robustness tradeoff in a general MDP model, which we believe is a first-of-its-kind result.
>
> While our main contribution lies in the theoretical side, we agree that empirical validation of our theoretical insights is essential to demonstrate the practical applicability and effectiveness of our approach. In the revised version of the paper, we will address this concern in the appendix by including empirical assessments of our approach. Furthermore, we will consider adding experiments to explore the impact of hyper-parameters, such as the tuning parameter $\beta$ in practice. Thank you again for your constructive feedback!

---

> > ### Comment · Reviewer_SGod · 2023-08-20
> > **Thanks for the rebuttal**
> >
> > Thank you for the rebuttal.
> >
> > I hope the authors will indeed include empirical assessment in the final revision of the paper, as this can be very valuable to the readers in general. I'd raise my rating to weak accept.

---

> > > ### Author Response · Authors · 2023-08-20
> > > **Thank you for your review**
> > >
> > > Thank you again for your thoughtful consideration and constructive feedback! We will incorporate more empirical results in the final version as suggested. We hope our responses have clarified your questions and we sincerely appreciate your willingness to raise the rating. Please let us know if you have any further comments!

---

### Official Review · Reviewer_BSCA · 2023-07-07

**Soundness:** 3 good
**Presentation:** 3 good
**Contribution:** 3 good
**Rating:** 7
**Confidence:** 3

**Summary:**

This paper investigates ways to combine untrusted learned policy and Q value predictions with a trusted baseline fixed policy such that the resulting combination achieves a higher performance bound and lower regret than established methods for combining the two. In particular, this paper studies how the use of Q-value predictions in addition to policy action predictions allows for stronger robust performance.

**Strengths:**

I'll first state that that my expertise is in the field of experimentally-driven deep reinforcement learning, so as this paper seems to come from the control-theory literature my ability to evaluate some aspects of it and to properly value its contribution to that field is limited. As such I will focus my comments on the elements where I can speak with some confidence and give the benefit of the doubt regarding other aspects (particularly the overall significance and originality of the problem and set of assumptions being made to the control theory community).

All that said, from my perspective, this paper makes a lot of sense- the core argument, that using information about TD error (or more generally the error estimates of the RL model- I can imagine there's ways to extend this work to other flavors of RL algorithms without Q-functions) allows for a better tradeoff between consistency and robustness, seems like a sensible one.

In deep RL it's common to use various measures of model error/confidence to constrain the actions and/or gradient updates of the model in a less rigorous way (Conservative Q Learning is a recent notable example of this), so doing the same thing to bound robustness guarantees makes sense to me.

Despite the difference in field, this paper communicated well and got the core arguments and algorithms across, and seems well written in general, though I can't comment on the perspective of a control theorist on this point.

**Weaknesses:**

To err on the side of generosity, I'm going to use this section to mostly speak to the practical feasibility and relevance of this paper (at least my sense of it). It's a theory paper, so I don't consider any of this a critical flaw in the work, but hopefully this is a useful perspective for the authors in their future work.

I'm unsure if assumption 3 is reasonable by default- it might require a very large Lipschitz constant for most neural network architectures (not an expert on this, but my understanding is most common architectures have very poor guarantees on curvature). While theoretically valid I worry that it might require the use of specialized models to get a reasonable constant in practice, especially since per equation 7 it looks like a specific value is needed for normalization, so there's incentive to use as small a number as possible.

The approximate TD error in equation 6 makes sense to use as an approximate trust measure, but I'll caution that in practice the magnitude of these values can vary by orders of magnitude and is highly MDP dependent, so there may be MDP/model dependent scale/normalization hyperparameters that need to be introduced. In theory and principle it makes sense to me, but my sense is that further work is needed to make this measure practical.

Another limitation might be applicability in contexts that are very different from those in which the Q-function was trained. Assuming the immediate reward/cost is not available or not highly informative (for example, if we have a lot of deferred and/or sparse rewards) and that the model is not being finetuned, in such cases the TD error between Q_t and Q_t+1 could be small despite the model being highly inaccurate in its Q estimates and thus its actions might be highly untrustworthy. "Confident but wrong" is a sadly common phenomenon in deep RL.

Overall though I don't think these issues are immediately relevant to the significance of this work, due to being both out of scope and possible to address through future work. As such (and being deferential regarding significance and novelty since I can't comment on those well) I'm inclined to recommend acceptance for this paper, subject to the opinions of other reviewers who may have perspective I lack.



**Questions:**

Thinking out loud, I wonder if relative TD error (compared to some MDP-specific baseline or a running average or similar) might make more sense in practice for equation 6 given the challenges introduced by deep RL models and MDP-specific scale variance. Relative TD error prioritization has been used for both prioritized experience replay and exploration, so there's evidence in deep RL that this sort of ranked/relative error measure is at least semi-reliable.

Regarding using various forms of structural information/error signals for RL, it might be useful for future work to look at some of the literature on exploration and robust/safety-conscious RL (if the authors are not already familiar with this literature) and how novelty/uncertainty metrics are used in those fields. A wide range of metrics have been proposed with differing tradeoffs which might be theoretically useful or at least provide inspiration.


**Limitations:**

While I can't speak well to the theoretical limitations of this paper, I have some discussion of possible practical limitations in the above sections (though such concerns are out of scope for this paper).

I don't see any potential negative societal impact from this paper. I will note the authors include a consideration of the potential impacts at the bottom of the paper regardless, which is pleasant to see.

---

> ### Author Rebuttal · Authors · 2023-08-09
>
> Thank you so much for the thorough review and the valuable insights provided. Your positive evaluation from a experimentally-driven deep RL perspective of the core argument regarding the derived tradeoff between consistency and robustness using TD error information is very encouraging.
>
> Regarding the concerns you raised about practical feasibility and relevance, we will consider adding some more concrete examples and empirical validations to demonstrate the efficacy of the proposed method. While our main focus is to understand theoretically how to go beyond the pure black-box setting and understand the fundamental tradeoff between consistency and robustness, we agree that the aspects mentioned by the reviewer are all important to consider. Let us address these points:
>
> 1. **Assumption 3**:
>
> >*I'm unsure if assumption 3 is reasonable by default- it might require a very large Lipschitz constant for most neural network architectures. While theoretically valid I worry that it might require the use of specialized models to get a reasonable constant in practice.*
>
> Guaranteeing a small Lipschitz constant can indeed be a practical concern for most neural network architectures. While we provided a theoretical basis for this assumption, we understand that in practice, obtaining a suitable constant $L_Q$ may be challenging and this depends on how the values functions are parameterized. To address this, we will discuss this limitation explicitly in the paper and highlight the need for further investigation into selecting or approximating the Lipschitz constant in practical scenarios.
>
> Besides, such a continuity assumption on the Q-value functions is common in the related literature. For example, similar assumption has been used to deal with the general continuous MDP in off-policy evaluation by using the Lipschitz property of the Q-functions [Tang et al.].
>
> It is worth mentioning that there is a line of works [Hinderer et al.], [Asadi et al.] that have studied the necessary Lipschitz smoothness conditions on the transition and reward functions, to make the value functions Lipschitz. Besides these conditions, as shown in [Gelada et al.], a parameterized
> latent space model (called DeepMDP) can be used to guarantee the Lipschitz continuity of the Q-value functions, for a class of Lipschitz-valued policies.
>
> 2. **Approximate TD error**:
>
> >*The approximate TD error in equation 6 makes sense to use as an approximate trust measure, but I'll caution that in practice the magnitude of these values can vary by orders of magnitude and is highly MDP dependent.*
>
> We recognize that the magnitude of approximate TD error values can vary significantly across MDPs and models. Your point about the necessity of introducing MDP/model dependent scale/normalization hyperparameters makes a lot of sense from a practical perspective.
>
> Our results validate the existence of such hyper-parameters to guarantee the stated consistency and robustness tradeoff. We will supplement empirical results in the appendix to empirical address this issue. Exploring adaptive scaling methods or learning these hyper-parameters alongside the algorithm can be interesting future directions.
>
> 3. **Applicability in Different Contexts**:
>
> >*"Confident but wrong" is a sadly common phenomenon in deep RL.*
>
> Thank you for bringing up this point. Our theoretical analysis may shed some light on alleviating the "confident but wrong" phenomenon in deep RL. Indeed, it can be the case that a value-based policy generates low TD-error while having bad performance in practice. The choice of the estimated TD-error used in the definition of $R_t$ has an interesting decomposition shown in Lemma 4 (Appendix B.3). This lemma is further used in the robustness analysis in Appendix D to show that because of a telescoping property, when such an accumulated estimated TD-error is small, then the expected dynamic regret is also small, utilizing Theorem D.1. In practice, using such an accumulated estimated TD-error with the right hyper-parameter $\beta$ can reflect the true performance of the value-based policy.
>
> >*I wonder if relative TD error (compared to some MDP-specific baseline or a running average or similar) might make more sense in practice for equation 6. It might be useful for future work to look at some of the literature on exploration and robust/safety-conscious RL and how uncertainty metrics are used in those fields.*
>
> It would be very interesting to theoretically understand if the relative TD error can provide more benefits. We highly appreciate the suggestion of incorporating ideas from the exploration and robust/safety-conscious RL literature. Our current work is not dealing with hard constraints but we are looking forward to investigate how to use novelty/uncertainty metrics to address different tradeoffs and challenges. We understand that the current work might have limitations that are out of scope, but we are committed to addressing them in our future research. Once again, your expertise and perspective have been invaluable, and we remain open to further feedback.
>
> [Hinderer et al.] Hinderer, Karl. "Lipschitz continuity of value functions in Markovian decision processes." Mathematical Methods of Operations Research 62 (2005): 3-22.
>
> [Asadi et al.] Asadi, Kavosh, Dipendra Misra, and Michael Littman. "Lipschitz continuity in model-based reinforcement learning." International Conference on Machine Learning. PMLR, 2018.
>
> [Gelada et al.] Gelada, Carles, et al. "Deepmdp: Learning continuous latent space models for representation learning." International Conference on Machine Learning. PMLR, 2019.
>
> [Tang et al.] Tang, Ziyang, et al. "Off-policy interval estimation with lipschitz value iteration." Advances in Neural Information Processing Systems 33 (2020): 7887-7897.
>
> [Kim et al.] Kim, Jeongho, and Insoon Yang. "Hamilton-Jacobi-Bellman equations for Q-learning in continuous time." Learning for Dynamics and Control. PMLR, 2020.

---

> > ### Comment · Reviewer_BSCA · 2023-08-11
> > **Response to Rebuttal**
> >
> > Glad my comments were useful!
> >
> > I'm glad to hear of your interest in developing this topic further- I think this is a good foundation that could be high-impact with further development and the resolution of practical details in future work. I appreciate your comments on feasibility, I agree these issues (and others that I can't anticipate offhand) should be surmountable, although I expect that to be enough of a project to be worth a paper in its own right.
> >
> > The broader topic of trust metrics (for various ends) is a complex but important one in deep RL IMO. A little further afield from this work, policy-gradient trust methods like Proximal Policy Optimization and Trust Region Policy Optimization constrain gradient updates based on how far they deviate from a prior policy, and have become widespread on-policy policy gradient method baselines and might spark inspiration if you aren't familiar with that literature.
> >
> > As a casual thought, since prediction error/trust/etc metrics are often useful for computing gradient updates, I wonder if there is some theoretical connection between the quality of a given metric for gradient updates and its quality for this sort of gray-box learning-augmented approach? At the very least computed TD-error variance and sampling bias both reduce gray-box trust and produce worse/slower-to-converge gradients. As an outsider to the topic, I could see that connection being relevant for online training of learning-augmented algorithms, where a policy is trusted very little initially and given more influence as training progresses.

---

> > > ### Author Response · Authors · 2023-08-15
> > >
> > > Great thanks for your invaluable insights and suggestions! The idea of exploring a wider scope of error/trust metrics (e.g. KL divergence in TRPO), especially as they relate to policy gradient methods like PPO or TRPO, is deeply thought-provoking. Venturing further, any such extension would likely necessitate integrating various "grey-box advice modalities", given our existing guarantees mainly work for value-based policies. Recognizing this, we're motivated to examine broader classes of policies.
> > >
> > > Besides, we’re particularly intrigued by your prospect of error/trust metrics in deep RL. The potential to unveil a connection between learning-augmented algorithm tradeoffs and policy gradient updates is highly interesting. For instance, it'd be interesting to characterize a tradeoff for a gradient-based learning-augmented policy that updates its policy based on the KL divergence such as ${D_{KL}}^{\rho_{\theta_{\text {old}}}}\left(\theta_{\text {old }}, \theta\right)$ in TRPO. Our current combination is similar to the policy mixture in the conservative policy iteration update [Kakade et al.] and the link you've pointed out between the gradient update's convergence quality and the consistency/robustness tradeoff offers a promising trajectory for future theoretical exploration. Our future direction leans towards devising a unified framework that may extend to both policy gradient updates (such as TRPO) and more practical error/trust metrics.
> > >
> > > Thank you again for your thoughtful engagement! If you have any additional insights or suggestions, please do share them — your expertise significantly enriches our journey!
> > >
> > >
> > > [Kakade et al.] Kakade, Sham and Langford, John. Approximately optimal approximate reinforcement learning. In ICML, volume 2, pp.
> > > 267–274, 2002.

---

### Official Review · Reviewer_9C69 · 2023-07-20

**Soundness:** 3 good
**Presentation:** 3 good
**Contribution:** 2 fair
**Rating:** 5
**Confidence:** 1

**Summary:**

This paper provides a theoretical study of the tradeoff between consistency and robustness in Markov Decision Process for which machine learned advice is available. With extensive theoretical analyses, the paper leverages Q-value advice for improving the impact of machine-learned advice in these contexts. Several hyperparameters are suggested which control the impact of these learned advices. The paper provides derivations for both black-box and grey-box settings.


I have read the rebuttal. I am more positive about this submission after the rebuttal and reading the other reviews.The authors have addressed my concerns regarding empirical validation, and the lack of practical examples.

**Strengths:**

- The paper studies an interesting and salient problem and provides theoretical derivations for their findings.
- The paper is clearly structured.
- Limitations and broader impact are adequately studied.
- Supplementary material is impressively extensive and detailed.

**Weaknesses:**

- I believe the paper could strongly benefit from empirical validation to better communicate the potential impact of their theoretical findings.
- Some decision choices (see Questions) require further discussion.
- This paper is at times hard to follow and could be written in a more clear way.

**Questions:**

- L 308: What is the impact of this approximation?
- L 283: Could you give a practical example of the impact of this radii?

**Limitations:**

Limitations and the broader impact of this method are properly addressed.

---

> ### Author Rebuttal · Authors · 2023-08-07
>
> Thank you for your thoughtful comments! Below are our responses to your specific comments and questions:
>
> 1. **Empirical Validation**:
>
> While our primary focus and main contribution center around the theoretical aspects, we fully agree with your suggestion that empirical validation can significantly enhance the manuscript's impact and better communicate the potential implications of our theoretical findings.
>
> In the revised manuscript, we will add some empirical results in the appendix to further demonstrate the applicability of our methods in real-world settings.
>
> 2. **Impact of Approximation**:
>
> >*What is the impact of this approximation?*
>
> Our algorithm relies on the estimated TD-error in equation (6). In our approach, the "robustness budget" $R_t$ is set according to equation (7), utilizing the estimated TD-error. This is because computing the true TD-error requires knowledge of the transition probabilities in the MDP, which is often not feasible.
>
> Throughout our analysis, we select $R_t$ as specified in equation (7). To handle the approximation error in our analysis, we introduce an additional technical lemma (Lemma B.3). of the supplementary material, which plays a crucial role in accounting for approximation errors. The key idea is to decompose the estimated TD-error into two terms representing the differences between Q-values. The proof is given in Appendix B.
>
> 3. **Practical Examples**:
>
> >*Could you give a practical example of the impact of this radii?*
>
> Below we use two real-world examples to demonstrate the impact of the radii (line 283).
>
> *A. Voltage Control.* Consider a practical scenario where we need to control an inverter in a power system to regulate voltage. The control action involves adjusting the reactive power. Considering the injections from renewable resources, the control dynamics can be formulated as an MDP.
>
> Recent research has shown promising results in using learning-based methods, such as RL or multi-agent RL, to control inverters in power systems [Marot et al.], [Wang et al.]. However, a major challenge arises when applying these methods to real systems, as they may suffer performance degradation when the environment changes, such as seasonal variations, human behaviors, policy changes, or extreme weather conditions. Safe RL methods [Sootla et al.] or specific neural network designs [Chen et al.] can provide guarantees of feasibility, but they may still fall short of optimal performance during distribution shifts, particularly when the MDP becomes non-stationary.
> Existing baseline methods, such as droop control, have been used for voltage and frequency control problems in power systems. However, these methods may not fully leverage the benefits of learning-based approaches.
>
> Our proposed methods offer a potential solution to bridge this gap. We can regulate the flexibility of the true actions to align with machine-learned actions centered at the Wasserstein robust actions. A larger radius indicates greater trust in the machine-learned actions, allowing the inverter control to follow the RL action and vice versa. By tuning the radii and finding the optimal balance between robustness and consistency, we can enhance the efficiency and reliability of inverter control in the face of dynamic and uncertain power system environments.
>
> *A. Autonomous vehicle Navigation.* Another example is in the context of autonomous driving. Imagine an autonomous vehicle navigating through a busy city. The vehicle's decision-making system relies on a mix of a pre-trained value-based policy and a baseline navigator based on external sources (e.g., traffic lights and pedestrian movements from the vehicle sensors).
>
> In this scenario, a small radius would mean the autonomous vehicle relies heavily on the baseline signals and is less willing to trust the machine-learned policy. This could lead to a safer but conservative plan, where the vehicle may miss opportunities to navigate through traffic more efficiently, especially during rush hours when traffic patterns are complex. On the other hand, a larger radius would imply the autonomous vehicle is more willing to trust the value-based policy .
>
> Choosing an appropriate radius implies a delicate balance. If the radius is too small, the vehicle may not fully utilize valuable action from the value-based policy, missing opportunities for efficient navigation. If the radius is too large, it may rely excessively on potentially untrustworthy advice, compromising the reliability. By characterizing how Q-value advice can improve the consistency and robustness tradeoff, the radii determined by the approximate TD-error offer provable solutions to strike the right balance. Optimally setting these radii ensures the autonomous vehicle can effectively integrate its own learned policy and the baseline signals, leading to safer, more efficient, and reliable autonomous driving behavior. We will clarify in the revised manuscript using these concrete applications to improve the readability.
>
> 4. **Paper is Hard to Follow**. Thank you for bringing out this issue! We will improve the readability of the paper by furnishing more practical examples as aforementioned and provide more and better explanations.
>
> **References**
>
> [Marot et al.] Marot, Antoine, et al. "Learning to run a power network challenge: a retrospective analysis." NeurIPS 2020 Competition and Demonstration Track. PMLR, 2021.
>
> [Wang et al.] Wang, Jianhong, et al. "Multi-agent reinforcement learning for active voltage control on power distribution networks." Advances in Neural Information Processing Systems 34 (2021): 3271-3284.
>
> [Sootla et al.] Sootla, Aivar, et al. "Sauté rl: Almost surely safe reinforcement learning using state augmentation." International Conference on Machine Learning. PMLR, 2022.
>
> [Chen et al.] Chen, Yize, et al. "Optimal Control Via Neural Networks: A Convex Approach." International Conference on Learning Representations. 2019.

---

> > ### Comment · Reviewer_9C69 · 2023-08-16
> > **Response to rebuttal**
> >
> > Thank you for your detailed response! The practical examples provided by the authors are very insightful. I am more positive about this submission after the rebuttal and the excellent analyses done by my fellow reviewers. I strongly encourage the authors to include, as promised, the empirical validation on the supplementary material.

---

> > > ### Author Response · Authors · 2023-08-16
> > >
> > > Thank you once more for your encouraging feedback! We're pleased that our explanations have clarified some of the raised concerns. Please do let us know if you have any further questions or concerns. Your insights are always valued!

---

### Official Review · Reviewer_c87H · 2023-07-26

**Soundness:** 3 good
**Presentation:** 3 good
**Contribution:** 3 good
**Rating:** 7
**Confidence:** 2

**Summary:**

Authors study algorithms for Marcov Decision Processes receiving advice in the
form of a value-based machine-learned policy. Value-based policies decide
their actions based on minimizing an estimated Q-function, which predicts
the best possible outcome in the rest of the time horizon given a current
action. Such policies are known to perform well in some practical scenarios,
but collapse in others.

They propose algorithms for two regimes. In the first regime, the algorithm
receives only the action taken by the value-based policy while, in the
second regime, it also receives an estimate of the Q-function used by the
value-based policy to determine its action.
In the first regime, they propose a consistency-robustness tradeoff.
In the second regime, they use the additional information (Q-function) to
estimate the quality of the advice provided by the value-based policy
and tune the trade-off parameter online, obtaining consistency 1 and
near-optimal robustness.
They show that the analysis of their algorithm for the first regime is tight.


**Strengths:**

* work on an important problem
* online tuning of the consistency-robustness trade-off
* tight analysis of their algorithm

**Weaknesses:**

* It is not clear whether consistency-robustness tradeoff with weaker predictions is necessary for any algorithm. They prove a lower bound only for their algorithm.
* requires machine-learned advice in a form of a value-based policies. Other kinds of policies might be also beneficial.

**Questions:**

* In algorithm 1, should line 6 be part of the for loop?

**Limitations:**

their ML-augmented algorithms are limited to a certain kind of predictions. Authors have mentioned this limitation clearly in the last section of their submission.

---

> ### Author Rebuttal · Authors · 2023-08-09
>
>
> Thank you so much for the valuable feedback on our work!  Below are our responses to the comments:
>
> 1. **Consistency-Robustness Tradeoff for Black-Box Setting**:
>
> >*It is not clear whether consistency-robustness tradeoff with weaker predictions is necessary for any algorithm. They prove a lower bound only for their algorithm.*
>
> We appreciate a lot your thoughtful observation regarding the necessity of the consistency-robustness tradeoff with weaker predictions for any black-box algorithm, as we mentioned in the limitation.
>
> The lower bound we have is for a special black-box procedure, which convexly combines actions given by the machine-learned policy and the robust baseline. Although this form is special, it has been widely used to design controllers such as in references [16] and [35] in the manuscript.
>
> Our main results show that additional "grey-box" advice (Q-values) can indeed be used to nontrivially improve the consistency and robustness tradeoff such that with the grey-box advice, we can actually do better than the best that we can hope for in the considered black-box setting, i.e., better than the lower bound in the black-box setting.
>
> We acknowledge that this point might not have been explicitly articulated in the paper, and we will revise the manuscript accordingly to provide clearer context.
>
> 2. **Beyond Value-Based Policies**:
>
> >*It requires machine-learned advice in a form of a value-based policies. Other kinds of policies might be also beneficial.*
>
> Thank you for bringing up the concern!
>
> It's important to note that our findings provide insights into the potential benefits of going beyond the pure black-box setting by incorporating other forms of advice in decision-making processes, resulting in near-optimal performance guarantees in the general MDP setting. Our current results focus on the specific case of machine-learned advice in the form of value-based policies, specifically Q-value advice.
>
> Despite this limitation, the class of value-based policies is still an important type of policies to consider. We chose this setting to investigate the tradeoff between consistency and robustness to highlight the advantages of leveraging additional information about how the advice is generated. While our current analysis focused on value-based policies, the implications of our work extend to the broader understanding of optimally using machine-learned advice for decision-making tasks.
>
> Currently, we acknowledge that there are other forms of policies that could be beneficial and might influence the consistency-robustness tradeoff differently. For example, policy-gradient-based approaches, model-based policies (knowledge of the model could be some grey-box information), or even ensemble methods could be relevant alternatives that deserve exploration in future research. As future works, it would be interesting to fully understand the impact of information from machine-learned algorithms by exploring and unifying  other forms of advice.
>
>
> 3. **Line 6 in Algorithm 1**:
>
> >*In algorithm 1, should line 6 be part of the for loop?*
>
> Thank you so much for pointing out this typo! We have fixed it in the revised manuscript.

---

> > ### Comment · Reviewer_c87H · 2023-08-12
> >
> > thank you for your explanation.

---

### Decision · Program_Chairs · 2023-09-21

**Decision:**

Accept (poster)

**Comment:**

The paper focuses on the important problem of online tuning of the consistency-robustness trade-off.
Some motivation of the proposed setting should be added, and the paper lacks empirical verification.
However, the theoretical contribution is reckoned to be significant, with many reviewers solidly recommending acceptance.